# The HDL particle composition determines its antitumor activity in pancreatic cancer

Raimund Oberle[1], Kristina Kührer[1], Tamina Österreicher[1], Florian Weber[2], Stefanie Steinbauer[3], Florian Udonta[4], Mark Wroblewski[4], Isabel Ben-Batalla[5], Ingrid Hassl[1], Jakob Körbelin[6], Matthias Unseld[7], Matti Jauhiainen[8], Birgit Plochberger[2], Clemens Röhrl[3], Markus Hengstschläger[1], Sonja Loges[9], Herbert Stangl[1]

Despite enormous efforts to improve therapeutic options, pancreatic cancer remains a fatal disease and is expected to become the second leading cause of cancer-related deaths in the next decade. Previous research identified lipid metabolic pathways to be highly enriched in pancreatic ductal adenocarcinoma (PDAC) cells. Thereby, cholesterol uptake and synthesis promotes growth advantage to and chemotherapy resistance for PDAC tumor cells. Here, we demonstrate that high-density lipoprotein (HDL)–mediated efficient cholesterol removal from cancer cells results in PDAC cell growth reduction and induction of apoptosis in vitro. This effect is driven by an HDL particle composition–dependent interaction with SR-B1 and ABCA1 on cancer cells. AAV-mediated overexpression of APOA1 and rHDL injections decreased PDAC tumor development in vivo. Interestingly, plasma samples from pancreatic-cancer patients displayed a significantly reduced APOA1-to-SAA1 ratio and a reduced cholesterol efflux capacity compared with healthy donors. We conclude that efficient, HDL-mediated cholesterol depletion represents an interesting strategy to interfere with the aggressive growth characteristics of PDAC.

## Introduction

In contrast to the well-studied role of high-density lipoproteins (HDL) in cardiovascular research, its functional impact on cancer biology is less clearly defined. Clinical investigations elaborated on the association of plasma apolipoprotein A1 (APOA1)/HDL levels and the risk of developing cancer, whereby the large majority of the studies reported an inverse association. For example, in randomized controlled trials of lipid-altering interventions, a significant inverse correlation between HDL cholesterol (HDL-C) and cancer incidence was found (1). Moreover, within the European Prospective Investigation into Cancer and Nutrition, the concentrations of HDL and APOA1 were inversely associated with the risk of colon cancer (2). In agreement with clinical data, preclinical studies that explored the mechanistic role of HDL in carcinogenesis predominantly attributed tumor protective functions for these lipoprotein particles. For example, B16F10 melanoma-bearing mice expressing a human APOA1 transgene exhibited reduced tumor burden, decreased tumor-associated angiogenesis, lower metastatic potential, and enhanced survival. These effects were reproduced by the injection of plasma-purified human APOA1 protein into *ApoA1* KO mice (3). To examine a causal role of reduced APOA1/HDL levels in patients suffering from ovarian cancer, mouse in vivo studies with ID8 ovarian adenocarcinoma cells revealed a significant antitumor capacity of the human *Apoa1* transgene and the therapeutic administration of APOA1 mimetic peptides (4). APOA1 and APOA1 mimetic peptides directly reduced the viability and proliferation of ID8 tumor cells and *cis*-platinum–resistant human ovarian cancer cell lines by the binding and removal of the mitogenic lipid lysophosphatidic acid (4).

Interestingly, a work by Cedo et al challenged the antitumor activity of mature, APOA1-containing HDL. By using a model of inherited breast cancer, transgenic overexpression of human APOA1 did not result in inhibition of tumor growth. In contrast, the APOA1 mimetic peptide D-4F significantly increased tumor latency and reduced tumor outgrowth (5). Of note, APOA1 mimetic peptides and discoidal reconstituted HDL, which mimic pre-β HDL particles in the circulation, are highly efficient acceptors of ATP-binding

[1]Center for Pathobiochemistry and Genetics, Medical University of Vienna, Vienna, Austria   [2]School of Medical Engineering and Applied Social Sciences, University of Applied Sciences Upper Austria, Linz, Austria   [3]Center of Excellence Food Technology and Nutrition, University of Applied Sciences Upper Austria, Wels, Austria   [4]Department of Oncology, Hematology and Bone Marrow Transplantation, University Comprehensive Cancer Center Hamburg, University Medical Center Hamburg-Eppendorf, Hamburg, Germany   [5]Division of Personalized Medical Oncology (A420), German Cancer Research Center (DKFZ), Heidelberg, Germany   [6]ENDomics Lab, Department of Oncology, Hematology and Bone Marrow Transplantation, University Medical Center Hamburg-Eppendorf, Hamburg, Germany   [7]Department of Medicine I, Division of Palliative Medicine, Medical University of Vienna, Vienna, Austria   [8]Minerva Foundation Institute for Medical Research and Finnish Institute for Health and Welfare, Genomics and Biobank Unit, Biomedicum 2U, Helsinki, Finland   [9]Department of Personalized Oncology, University Hospital Mannheim, Medical Faculty Mannheim, University of Heidelberg, Mannheim, Germany

Correspondence: raimund.oberle@meduniwien.ac.at

cassette subfamily A member 1 (ABCA1)–mediated cholesterol ef-flux (6, 7, 8). In contrast, spherical, lipid-rich mature HDL particles serve as high affinity ligands and donors for bidirectional, scavenger receptor class B type 1 (SR-B1)–mediated cholesterol transport at the plasma membrane (9).

Pancreatic cancer is one of the deadliest and least therapeu-tically approachable malignancies with a median 5-yr survival rate of ~5–10% (10, 11). The prime reasons for this dismal prognosis are difficulties in early diagnosis and a highly diverse and hostile tumor microenvironment (TME) that promotes therapy unresponsiveness and fast developing resistance mechanisms (10, 12). This hostile TME forces tumor cells to metabolically adapt to meet their specific requirements for proliferation, migration, and invasion. Tran-scriptomic analyses revealed that lipid metabolic pathways are enriched in pancreatic ductal adenocarcinoma (PDAC) compared with normal pancreas. In particular, cholesterol uptake processes such as low-density lipoprotein receptor (LDLR) expression are highly activated in the malignant tissue (13). The inhibition of cholesterol uptake by PDAC cells in turn was shown to reduce cancer cell proliferation and sensitizes these cells toward che-motherapeutic interventions, thereby identifying this metabolic axis as an interesting novel target for therapeutic applications (13). Interestingly, APOA1, the main structural component of HDL and well known for its capacity to remove excess cholesterol from peripheral cells, has been identified as a potential biomarker in the detection of PDAC by comparative serum protein expression pro-filing (14). Although a causative link between cholesterol depletion and a reduction in pancreatic-cancer malignancy seems likely, whether and how HDL particles exert an anti-tumorigenic effect in PDAC remains largely unknown.

By combining in vitro analyses that elucidate the impact of HDL particles on tumor cells with in vivo experiments using *Apoa1* KO mice, we show that the HDL particle composition likely determines its antitumor activity. Reconstituted, small discoidal HDL particles displayed an increased ability to block tumor cell growth compared with lipid-rich, cholesterol-laden HDLs. This anti-tumorigenic ca-pacity of small rHDL particles correlated with a lower affinity to SR-B1–mediated lipid influx and a higher affinity to ABCA1-mediated cholesterol efflux. This study provides evidence for a particle composition-based antitumor activity of HDL in PDAC, which is at least in part regulated by an efficient cholesterol acceptor function of small, lipid-poor HDLs.

# Results

### Cholesterol depletion and small, discoidal HDL particles efficiently inhibit the growth of PDAC cell lines

Previous studies indicate that cancer cells show increased sensi-tivity toward cholesterol depletion because of their high need of cholesterol for cellular growth (15). In pancreatic cancer, the blockade of cholesterol uptake and the depletion of cholesterol availability via statins have been shown to reduce pancreatic cancer risk in preclinical and clinical settings (13, 16). In accordance, reducing cholesterol availability by culture of cells in lipoprotein

deficient serum (2% LPDS) decreased the viability of murine pancreatic adenocarcinoma cells Panc02, and cholesterol deple-tion by lovastatin further reduced cellular viability (Fig 1A). By comparing pancreatic-cancer cell lines Panc02, 6066 (17, 18), and BxPC3 regarding their sensitivity toward cholesterol depletion, all three cell lines demonstrated reduced cellular viability, with Panc02 cells being the most sensitive (Fig 1B). HDL particles serve as important acceptors of cellular cholesterol, with the capacity to remove excess cholesterol from peripheral cells (19). Importantly, and within the HDL pool, cholesterol efflux capacity differs according to the HDL particle size, lipid and protein composition, and the specific affinities for cellular efflux receptors such as ABCA1, ABCG1, or SR-B1 (9, 20, 21). By comparing the impact of native human HDL with small, lipid-poor reconstituted HDLs (rHDL, predominantly cholesterol acceptors) on the proliferative capacity of Panc02 cancer cells, we observed that under serum starvation conditions, rHDLs reduced cellular viability more efficiently (Fig 1C). Although HDLs decreased Panc02 viability also with increasing concentra-tions of serum in the cell culture medium, the particle-specific effect was attenuated (Fig 1C). To evaluate a potential particle-specific antitumor effect of HDL, we produced rHDL particles with varying lipid compositions to either mimic small, discoidal HDL particles (rHDL1) or spherical-like, lipid-rich HDLs (rHDL2; Fig 1D). In contrast to native HDL particles, rHDL1 and rHDL2 showed distinct migration patterns when applied to 1% native agarose gel elec-trophoresis, with rHDL2 showing decreased electrophoretic mo-bility (Fig 1E). Atomic force microscopy (AFM) analysis confirmed a discoidal shape for rHDL1 particles with a mean height of 3.9 ± 2.6 nm, a mean width of 11.5 ± 6.3 nm, and an aspect ratio (AR, height/width × 100) of 33.9%, which fits well with other studies from the literature describing similar sizes for disc-shaped reconstituted rHDL (Fig 1F and G and (22, 23, 24)). Importantly, rHDL2 particles differ in size and shape from rHDL1. Those particles exhibited a mean height of 14 ± 7.7 nm and a mean width of 19.7 ± 7.0 nm when analyzed with AFM. With an AR of 71.7%, those particles acquire a more spherical character and show increased heterogeneity compared with the rHDL1 preparations (Fig 1H and I). Gas chro-matography (GC) analysis of the rHDL preparations revealed a ~20-fold enrichment of cholesterol and cholesteryl esters in the rHDL2 particles, which parallels the theoretical molar composition of the particles used in reconstitution experiments (Fig 1J and K).

### The HDL particle composition affects viability and apoptosis of PDAC cells

Treatment of Panc02 and 6066 murine PDAC cell lines revealed that small HDL discs (rHDL1) induced a profound reduction in cellular viability, whereas treatment with nHDL and rHDL2 resulted in only a mild attenuation in pancreatic-cancer cell viability (Fig 2A). As persistent cholesterol starvation can induce cellular apoptosis (25, 26), we analyzed apoptosis rates in HDL-treated pancreatic-cancer cells (representative FACS blots are shown in Fig 2B). In contrast to nHDL and rHDL2, rHDL1 treatment led to a significant reduction of the live cell population (Fig 2C). In addition, HDL treatment in-creased early apoptosis rates compared with control irrespective of the particle composition (Fig 2D). In agreement with the data from

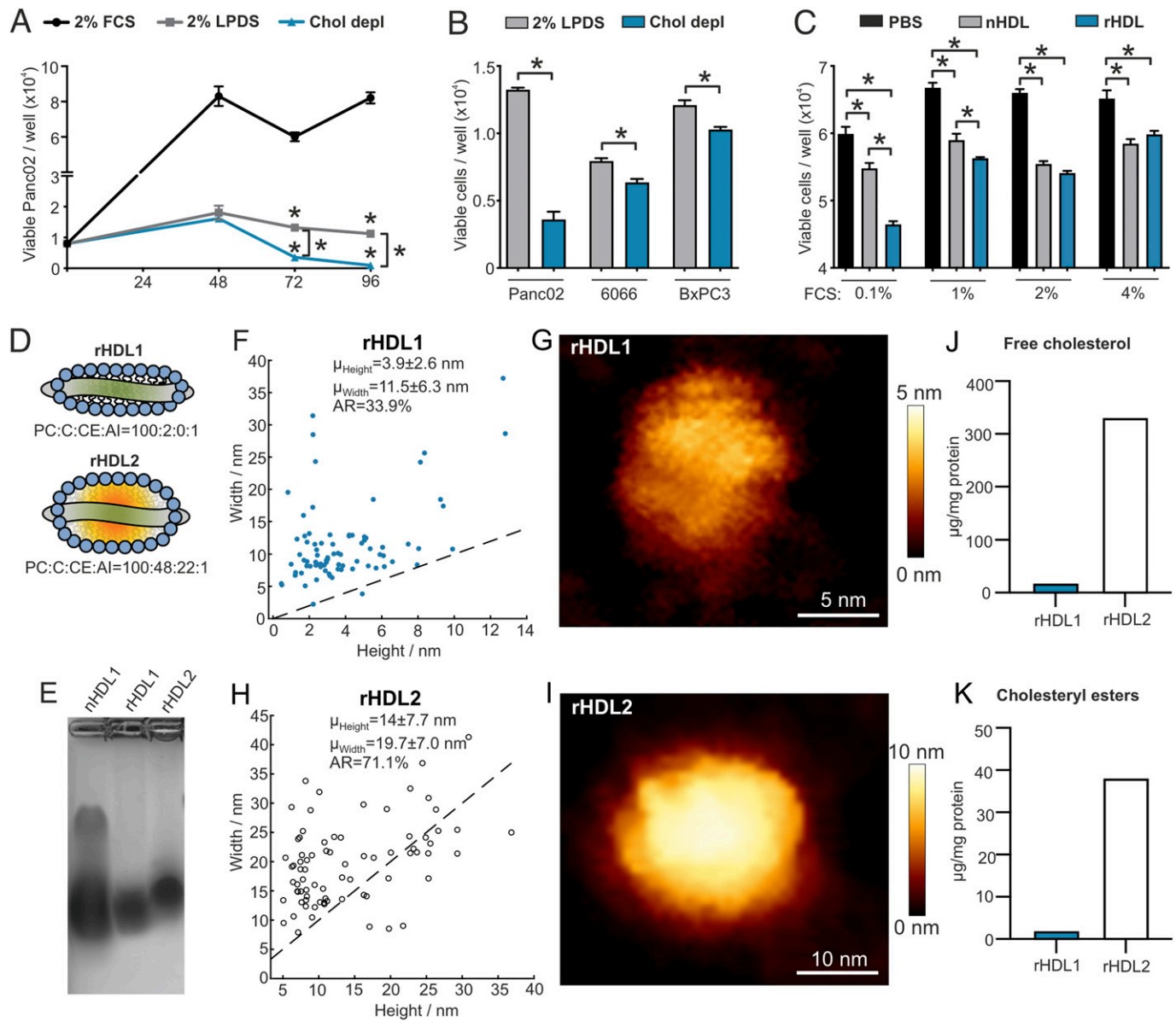

**Figure 1. Pancreatic-cancer cells are sensitive toward cholesterol depletion and rHDL particles.**
**(A)** Panc02 cells were grown in RPMI medium containing either 2% FCS, 2% LPDS, or 2% LPDS with the addition of 5 µM lovastatin and 100 µM mevalonate. Viability of cells was determined every 24 h using a WST1 viability assay (n = 4/4/4; *P < 0.05; one-way ANOVA). **(B)** Viability was determined of Panc02, 6066, and BxPC3 pancreatic-cancer cells after culture of cells for 72 h in either RPMI medium with 2% LPDS or cholesterol depleted medium (n = 4/4; *P < 0.05; unpaired t test). **(C)** Panc02 cells cultured in RPMI medium with increasing concentrations of FCS were treated for 48 h with PBS, high-density lipoprotein (HDL) isolated from human plasma of a healthy donor (75 µg/ml) or reconstituted HDL (75 µg/ml, molar ratio of PC:C:CE:APOA1 of 100:12.5:0:1, ZLB Behring), and viability was determined using a WST1 assay (n = 4/4/4; *P < 0.05; one-way ANOVA). **(D)** HDL particles were reconstituted using the indicated molar ratios of phosphatidylcholine (PC), cholesterol (C), cholesterol ester (CE), and apolipoprotein A-I (APOA1). **(E)** Electrophoretic mobility of HDL particles was tested on a 1% agarose gel. **(F)** Size distribution of rHDL1 particles measured by atomic force microscopy. **(G)** Representative deconvoluted image of an rHDL1 particle. **(H)** Size distribution of rHDL2 particles measured by atomic force microscopy. **(I)** Representative deconvoluted image of an rHDL1 particle. **(J, K)** Free cholesterol and cholesteryl ester content of rHDL1 and rHDL2 particles determined by gas chromatography.

viability assays (Fig 2A), rHDL1 was able to significantly expand the late apoptotic cell pool, indicating substantial cancer cell killing activity of this discoidal, lipid-poor HDL particle (Fig 2E).

The depletion of cellular cholesterol pools activates endogenous cholesterol synthesis pathways and LDLR expression to increase the exogenous uptake of cholesterol (27, 28). Therefore, we hypothesized that rHDL treatment might influence transcription of key enzymes of the cholesterol synthesis/uptake machinery. Indeed,

and in contrast to control-, nHDL-, and rHDL2-treated cells, rHDL1 particles significantly induced the expression of 3-hydroxy-3methyl-glutaryl-coenzyme A reductase (*Hmgcr*) at 50 and 75 µg/ml (Fig 2F; a list of murine oligonucleotide primers can be found in Table S1). Although rHDL1 and nHDL dose-dependently increased mRNA levels of the hydroxymethylglutaryl-CoA synthase (*Hmgcs*), rHDL2 failed to do so (Fig 2G). Furthermore, the lipid-poor rHDL1 particles increased *Ldlr* gene expression at 50 and 75 µg/ml (Fig 2H). Together, these data

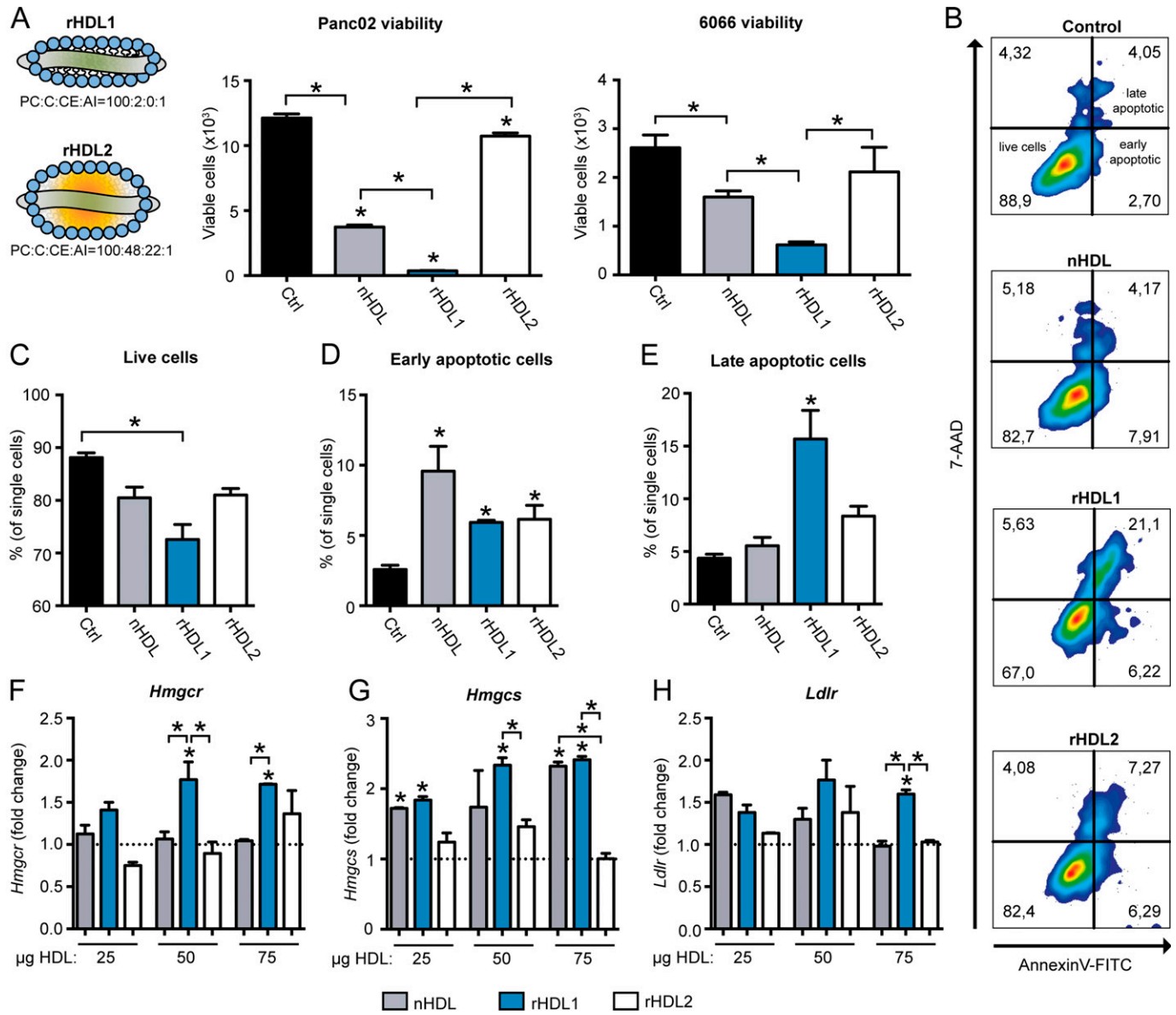

**Figure 2. The high-density lipoprotein (HDL) particle composition differentially affects pancreatic-cancer cell growth characteristics.**
**(A)** After a 24-h starvation period, Panc02 and 6066 cells were cultured in RPMI medium with 2% FCS in the presence of PBS, native human HDL (nHDL), or the indicated rHDL particles (75 μg/ml) for 48 h. Afterward, viability was determined using a WST1 assay (n = 4 biological replicates per group; *P < 0.05; unpaired t test). **(B)** Representative flow cytometry blots of Panc02 cells treated with different HDL particles (75 μg/ml) stained for the detection of apoptosis using 7-AAD and Annexin V. **(B, C, D, E)** Quantification of live, early apoptotic, and late apoptotic cells from flow cytometry experiments shown in (B) (n = 3 biological replicates per group, *P < 0.05; one-way ANOVA). **(F, G, H)** Panc02 cells were starved for 8 h and afterward treated with indicated concentrations of different HDL particles for 16 h. qPCR experiments show relative mRNA levels of *Hmgcr*, *Hmgcs*, and the *Ldlr*. Data are expressed as fold change over PBS-treated control cells (n = 3 replicates per group; *P < 0.05; one-way ANOVA).

indicate that rHDL1 particles reduce viability and induce apoptosis of pancreatic-cancer cells, paralleled by an induction of the cellular cholesterol synthesis and import machinery.

## The HDL particle composition determines its cholesterol efflux capacity from PDAC cells

These observed effects suggested efficient cholesterol depletion of cancer cells when treated with rHDL1 particles. To measure the cholesterol efflux capacity of different HDL species, we labeled

Panc02 and 6066 cells with [3]H cholesterol and analyzed the transfer of the radiotracer onto HDL. As anticipated from previous results, rHDL1 was able to remove significantly more cholesterol from cancer cells compared with nHDL. However, the cholesterol efflux capacity of lipid-rich rHDL2 particles even exceeded the one of rHDL1 (Fig 3A). Cholesterol efflux is primarily regulated by cell surface receptors such as the ABC-transporters ABCA1 and ABCG1 and the HDL receptor SR-B1. Whereas expression levels of *Scarb1* are high in Panc02 cells, *Abca1* expression is rather low and *Abcg1* mRNA levels are hardly detectable (Fig S1). Therefore, and to get a

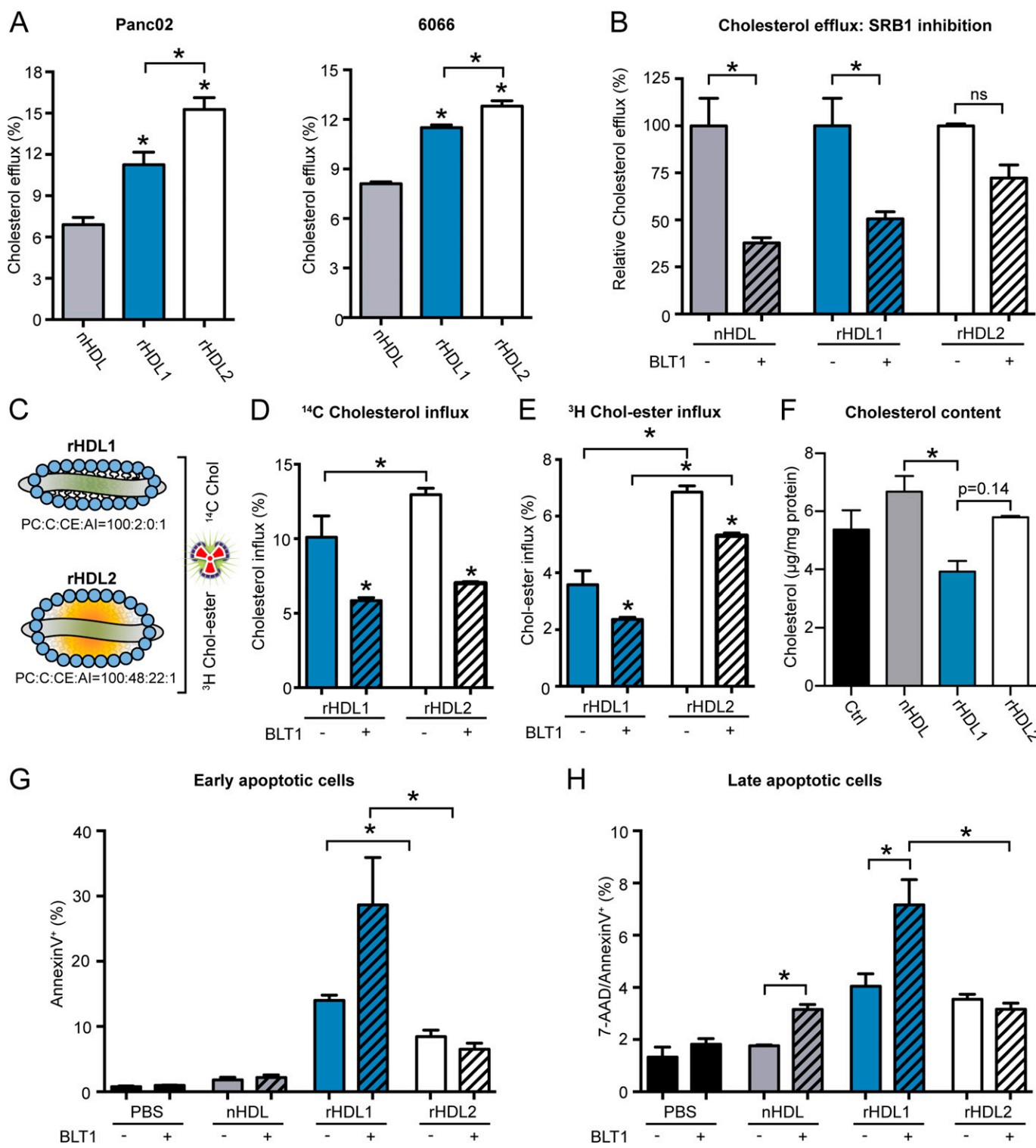

**Figure 3. SR-B1 influences cholesterol efflux and antitumor properties of rHDL particles.**
**(A)** [3]H cholesterol-loaded Panc02 and 6066 cells were subjected to cholesterol efflux assays with indicated high-density lipoprotein (HDL) particles (10 µg/ml) for 8 h. Cholesterol efflux is shown as % of transferred tracer from cells to HDL particles compared with control conditions (n = 4; *P < 0.05; one-way ANOVA). **(B)** relative, particle-specific cholesterol efflux capacity was analyzed in the absence or the presence of BLT1 (n = 4; *P < 0.05; one-way ANOVA). **(C)** Schematic representation of tracer-labeled rHDL particles. **(D, E)** [14]C cholesterol and [3]H cholesteryl ester influx, respectively, from rHDL to Panc02 cells in the absence or presence of BLT1(n = 4; *P < 0.05; one-way ANOVA). **(F)** Cellular cholesterol content was measured by GC in Panc02 cells treated for 48 h with different HDL species (75 µg/ml) in the presence of 2% FCS and 5 µM lovastatin (n = 2, *P < 0.05; one-way ANOVA). **(G, H)** quantification of early and (H) late apoptotic cells upon treatment of Panc02 cells with indicated HDL particles in the absence or presence of BLT1 (n = 3; *P < 0.05; unpaired t test).

more detailed view of the cholesterol transport properties of the different HDL particles, we first analyzed cholesterol efflux in the presence or absence of the SR-B1–blocking small-molecule inhibitor BLT1 (29). Pretreatment of Panc02 cells with BLT1 significantly reduced cholesterol efflux to nHDL and rHDL1 particles. In contrast, efflux of [3]H cholesterol toward rHDL2 particles was only moderately affected (Fig 3B). As SR-B1 mediates bidirectional lipid transfer (9), we synthesized rHDL particles containing both [3]H cholesteryl ester and [14]C cholesterol (Fig 3C) and used those particles to analyze lipid influx by measuring the accumulation of intracellular radiotracer molecules. Thereby, the influx of free cholesterol was significantly increased when Panc02 cells were incubated with the lipid-laden rHDL2 particles. BLT1-mediated SR-B1 inhibition blunted the influx of free cholesterol to a similar extent (Fig 3D). Interestingly, lipid-laden rHDL2 particles substantially exceeded rHDL1 particles in their ability to transfer cholesteryl oleate onto pancreatic-cancer cells. Similar to the data observed for cholesterol efflux, BLT1 blocked this effect only to a small extent (Fig 3E). These data identify rHDL2 particles to be more efficient in mediating lipid exchange with pancreatic-cancer cells compared with lipid-poor rHDL1 particles and an overall reduced efficacy of BLT1 to block rHDL2-mediated lipid flux. Of note, GC analysis revealed a reduction in cellular cholesterol concentration upon treatment with rHDL1 particles, indicating net cholesterol depletion in Panc02 cells specifically by rHDL1 particles (Fig 3F). Apoptosis assays again confirmed the highest apoptosis-inducing capacity for rHDL1 particles. Interestingly, BLT1 treatment of Panc02 cells even further induced early and late apoptotic cells in the presence of rHDL1 but not of rHDL2 particles (Fig 3G and H). Possible explanations for these data might be the exacerbated depletion of cellular cholesterol pools in the presence of rHDL1 and BLT1 or the increased affinity of rHDL2 toward SR-B1, which would result in a reduced ability of BLT1 to block lipid transport. This phenomenon of high affinity of lipid-rich HDLs toward SR-B1 has been previously described (reviewed in reference 9), which might further be exacerbated by the high expression levels of SR-B1 in Panc02 cells.

### The LXR agonist TO901317 increases rHDL1-specific cholesterol efflux and apoptosis

The rHDL1-specific increase in apoptosis upon inhibition of SR-B1 might also indicate the involvement of SR-B1-independent mechanisms that mediate the antineoplastic effect of rHDL1. As small and lipid-poor HDL particles are highly efficient acceptors of ABCA1-mediated cholesterol efflux, we analyzed the potential role of ABCA1 in the cholesterol efflux-driven antiproliferative effects of rHDL1 particles. To manipulate ABCA1 protein levels in Panc02 cells, which only show low endogenous expression levels of this protein, we used the LXR agonist TO901317 (TO, Fig 4A). As expected, TO treatment led to a 35% increase of cholesterol efflux to rHDL1 particles. Although cholesterol efflux to rHDL2 particles was also increased in the presence of TO, this effect was significantly weaker compared with rHDL1-mediated efflux (Fig 4B). Importantly, only rHDL1 particles reduced the amount of intracellular [3]H cholesterol in the presence of TO, pointing toward an involvement of ABCA1 in the rHDL1-mediated depletion of cellular cholesterol pools in Panc02 cells (Fig 4C). Finally, SR-B1 inhibition of TO-treated, ABCA1-expressing cells showed a

profound pro-apoptotic effect on Panc02 cells in the presence of rHDL1 particles, which was blunted when rHDL2 particles were used (Fig 4D).

Together, lipid-poor, discoidal-like rHDL1 particles induced a significant pro-apoptotic effect in Panc02 cells by unidirectional cellular cholesterol removal via ABCA1. In contrast, the pro-apoptotic effect was diminished when lipid-rich rHDL2 particles or native HDL isolated from human plasma were used. The previously reported high affinity of those particles to SR-B1 and their increased efficacy in mediating bidirectional lipid flux reduces the net cholesterol-removing capacity of those particles, thereby making them less effective in killing pancreatic-cancer cells.

### Liver-specific adeno-associated viral (AAV)-mediated APOA1 expression and rHDL injections reduce tumor burden in Panc02-bearing *ApoA1* KO mice

To analyze a potential antitumor effect of HDL particles in vivo, we first compared tumor growth kinetics of WT and APOA1-deficient mice (*Apoa1* KO), which exhibit dramatically reduced plasma HDL levels (30). Of note, we were unable to detect significant differences in tumor growth and tumor weight in the Panc02 tumor model and in the B16F10, lung carcinoma cell (LLC), and E0771 tumor models, which demonstrates an insignificant antitumor effect of endogenous, murine HDL particles (Fig S2A–D). Next, and as a consequence of the data obtained from in vitro experiments, we decided to artificially introduce APOA1/rHDL particles into tumor-bearing *Apoa1* KO mice. Therefore, we expressed murine APOA1 in the liver of Panc02 tumor-bearing mice using adeno-associated viral particles (AAV-APOA1). Robust expression levels of the APOA1 mRNA were detected in the livers of end stage Panc02 tumor-bearing WT and *Apoa1* KO AAV-APOA1 mice, whereas APOA1 mRNA levels were absent from livers of *Apoa1* KO mice (Fig 5A). Western blot analysis revealed the absence of APOA1 from plasma samples of *Apoa1* KO mice, whereas APOA1 protein was readily detectable in the plasma 5 d post–AAV injection and further increased after 14 and 21 d (Fig 5B). AAV-mediated APOA1 expression significantly increased HDL-C levels compared with APOA1 deficient mice, although HDL-associated cholesterol levels clearly remained below those in WT mice (Fig 5C). Importantly, AAV-APOA1 expression in *Apoa1* KO mice significantly reduced Panc02 tumor growth kinetics and tumor weight at experimental end stage (Fig 5D and E). AAV-mediated expression of murine APOA1 in WT mice only moderately reduced intraperitoneal Panc02 tumor weight compared with mice receiving control AAVs (Fig 5F). In addition, and to examine a therapeutic potential of rHDL particles in vivo, we compared Panc02 tumor growth in WT mice, *Apoa1* KO mice receiving PBS, and *Apoa1* KO mice receiving intravenous injections of rHDL (0.2 mg per injection every other day). Thereby, rHDL reduced Panc02 tumor weight significantly compared with *Apoa1* KO mice, pointing toward a moderate antitumor effect of rHDL particles also in vivo (Fig 5G).

As APOA1 (mimetic peptides)/HDL has been previously demonstrated to affect tumor angiogenesis and tumor-associated immune-cell populations (3), we measured intratumoral hypoxia and immune cell infiltration. Thereby, we found that AAV-mediated APOA1 expression did not affect tumor-associated hypoxia, a surrogate marker for tumor oxygenation (Fig S3A and B). Flow cytometric analysis of tumor-associated immune-cell populations showed an increase in MRC1[+] M2-polarized macrophages in the *Apoa1*

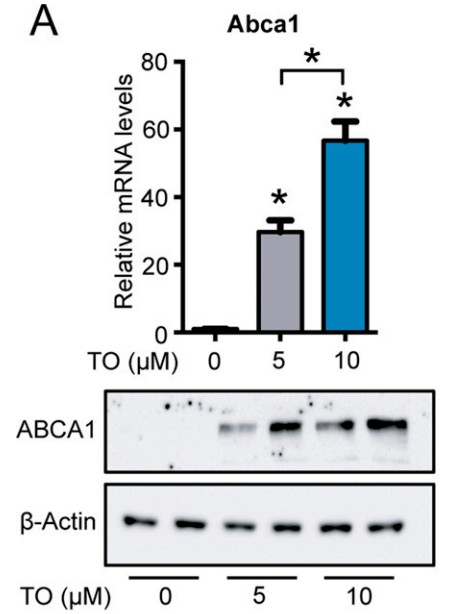

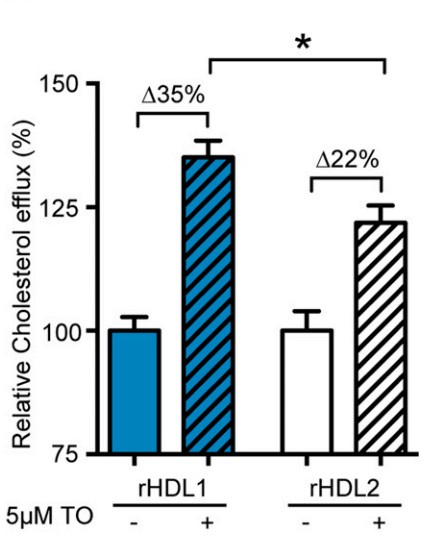

**Figure 4.  The LXR agonist TO901317 increases cholesterol efflux and apoptosis-inducing properties of small, lipid-poor rHDL.**
**(A)** TO901317 induces ABCA1 mRNA and protein levels in Panc02 cells (n = 3; *P < 0.05; one-way ANOVA).
**(B)** Relative, particle-specific cholesterol efflux from Panc02 cells in the absence or presence of TO901317 (n = 4; *P < 0.05; unpaired t test). **(C)** ³H cholesterol accumulation in Panc02 cells in the presence of TO901317 after efflux to indicated high-density lipoprotein particles (n = 4; *P < 0.05; one-way ANOVA).
**(D)** Relative apoptosis rates of TO901317-treated Panc02 cells induced by either rHDL1 or rHDL2 in the absence or presence of BLT1 (n = 3; P < 0.05; unpaired t test).

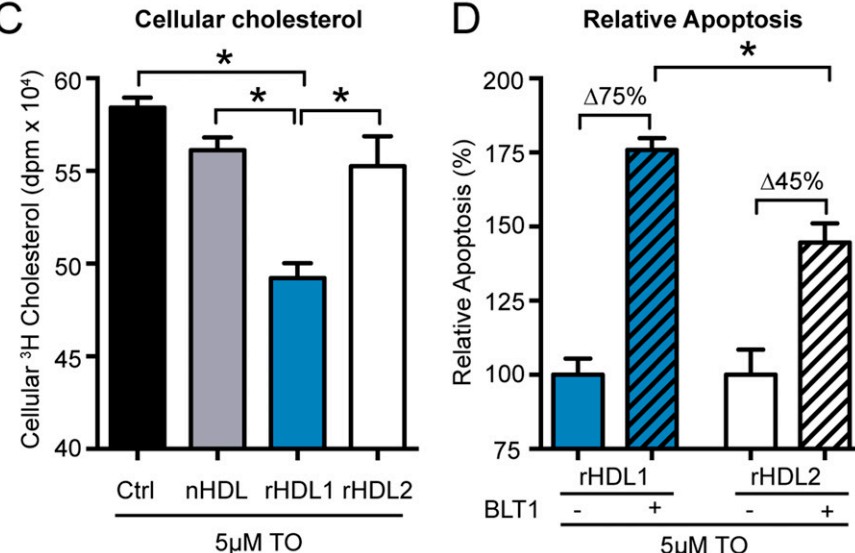

KO background, which was not substantially reverted by reintroduction of APOA1 (Fig S3C). Other tumor-associated immune-cell populations such as myeloid-derived suppressor cells (MDSC), CD8[+] cytotoxic T cells, and CD4[+] T-helper cells remained unchanged among all three groups of mice (Fig S3D and E). Therefore, we speculate that in parallel to the data from in vitro experiments, HDL particles mediate the observed anti-tumor effect in a direct manner. One possible explanation for a direct apoptosis-inducing effect of rHDL1 particles could be the disintegration of lipid rafts, which has been shown to reduce AKT phosphorylation, thereby also reducing cellular proliferation, eventually leading to the activation of pro-apoptotic mechanisms in cancer cells (31, 32, 33). By testing this hypothesis in our model, Western blot analyses demonstrated a reduction of AKT phosphorylation upon rHDL1 treatment,

whereas cholesterol-rich rHDL2 particles rather induced AKT phosphorylation (Fig 5H–J). In addition, rHDL1 particles efficiently induced the activation of caspases 3 and 7 in Panc02 and 6066 PDAC cell lines, thereby further substantiating the profound and direct-acting apoptosis-inducing activity of discoidal rHDL1 (Fig 5K and L).

### Decreased HDL-C and decreased efflux capacity of plasma samples from PDAC patients

To test the hypothesis that the HDL efflux capacity might be affected in the presence of tumor malignancies, we collected plasma samples from late-stage pancreatic-cancer patients. Although pancreatic-cancer plasma samples showed no difference in

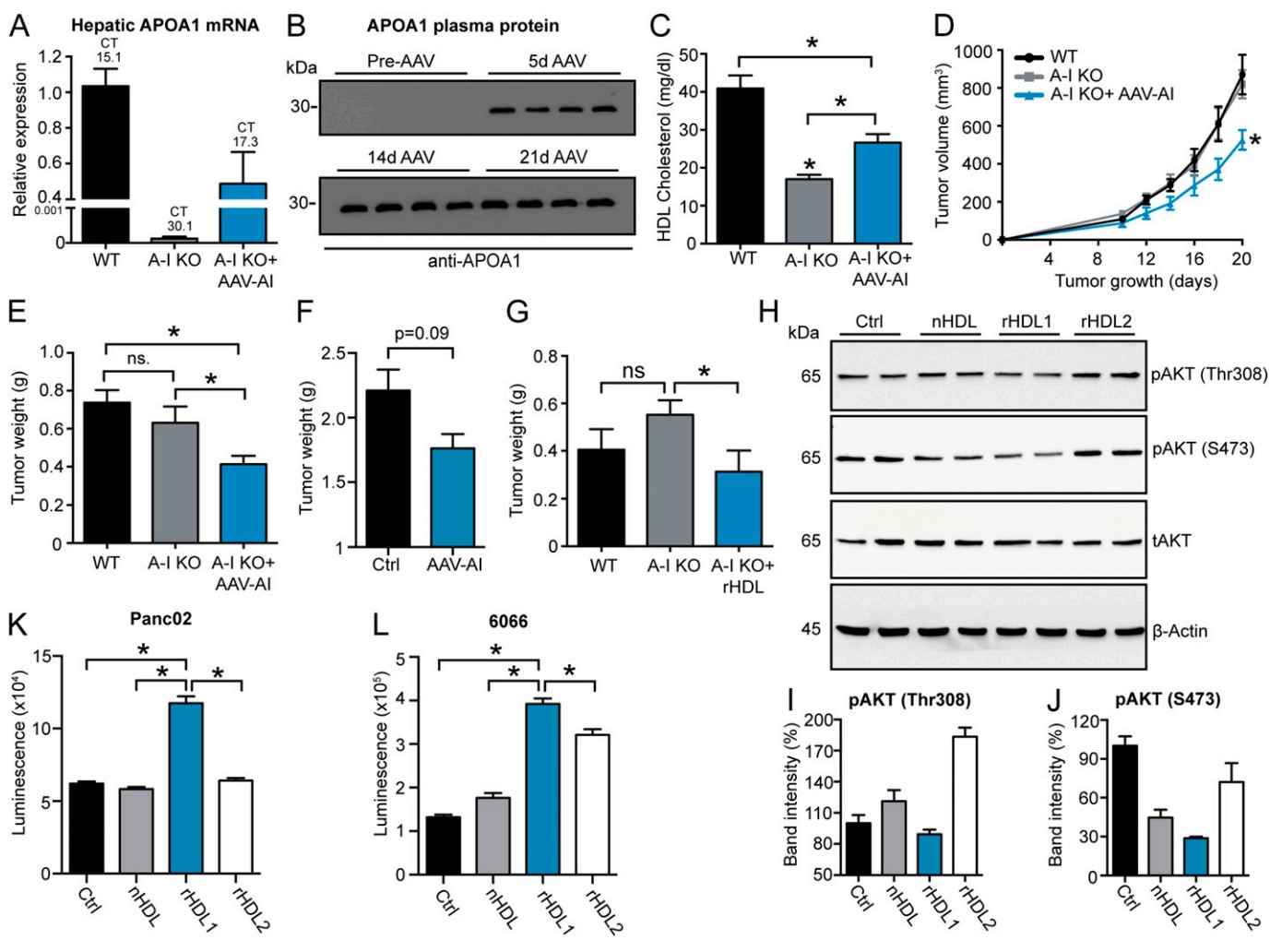

**Figure 5. AAV-mediated APOA1 reconstitution/rHDL injection reduces Panc02 tumor growth in *Apoa1* KO mice.**
**(A)** mRNA expression levels of APOA1 in liver tissue of Panc02-bearing WT, *Apoa1* KO, and AAV-APOA1–reconstituted *Apoa1* KO mice at experimental end stage (n = 8/6/7). **(B)** Western blot analysis of APOA1 protein levels in plasma of *Apoa1* KO mice prior and post AAV-APOA1 injection. **(C)** High-density lipoprotein (HDL)-C levels in Panc02 tumor-bearing mice at the experimental end stage (n = 8/5/7; *P < 0.05; one-way ANOVA). **(D, E)** Panc02 tumor growth kinetics and (E) tumor weight at experimental end stage (n = 8/5/7; *P < 0.05; unpaired *t* test). **(F)** Panc02 tumor weight in WT mice either receiving AAV-Ctrl or AAV-APOA1 particles (n = 8/4, unpaired *t* test). **(G)** Panc02 tumor weight of WT, *Apoa1* KO, and rHDL (0.2 mg per injection; PC:C:CE:APOA1 = 100:12.5:0:1, ZLB Behring)-injected *Apoa1* KO mice at an experimental end stage (n = 7/7/7; *P < 0.05; unpaired *t* test). **(H)** Western blot analysis of Panc02 cell lysates upon treatment of cells with indicated HDL particles (75 µg/ml) for 18 h. **(I, J)** Quantification of band intensities of pAKT (Thr308) and pAKT (S473), respectively, normalized to β-actin. **(K, L)** serum starved Panc02 or 6066 cells, respectively, were treated with indicated HDL particles (75 µg/ml) in the presence of 2% FCS for 24 h followed by the analysis of caspase 3/7 activation (n = 4; *P < 0.05; one-way ANOVA).

triglyceride and total cholesterol levels (Fig 6A and B), HDL-C levels were significantly decreased when compared with plasma samples of a cohort of healthy volunteers (Fig 6C). Of note, plasma samples of pancreatic-cancer patients exhibited decreased cholesterol efflux capacity compared with healthy control subjects (Fig 6D). This effect persisted when efflux was normalized to APOA1 plasma protein levels (Fig 6E). Interestingly, SAA1, an acute phase protein that associates with and potentially decreases the efflux capacity of HDL particles (34), was highly enriched in plasma samples of pancreatic-cancer patients, thereby reducing the APOA1-to-SAA1 ratio in those patients (Fig 6F). Stratification of pancreatic-cancer plasma samples into low (<13 µg/ml) and high (>13 µg/ml) SAA1 abundance revealed an inverse correlation with its cholesterol efflux capacity (Fig 6G). These data indicate a decreased HDL efflux

capacity in tumor patients, which might be explained in part by SAA1-mediated HDL remodeling.

In conclusion, our data indicate that the HDL cholesterol efflux capacity and therefore its antitumor activity is regulated by the HDL particle composition and/or the expression pattern of cell surface receptors such as SR-B1 and ABCA1 on tumor cells. Interestingly, in silico Kaplan–Meier analyses with the UCSC Xena database (35) neither revealed a correlation of patient overall survival with *SCARB1* (SR-B1) nor with *ABCA1* expression in the Genomic data commons (GDC) The Cancer Genome Atlas TCGA pancreatic-cancer cohort (Fig S4A and B). In contrast, further analyses revealed a potential dysregulation of cholesterol homeostasis in PDAC, showing an inverse association of *LDLR* and *HMGCS* expression levels with overall survival in this patient cohort (Fig 6H and I). This

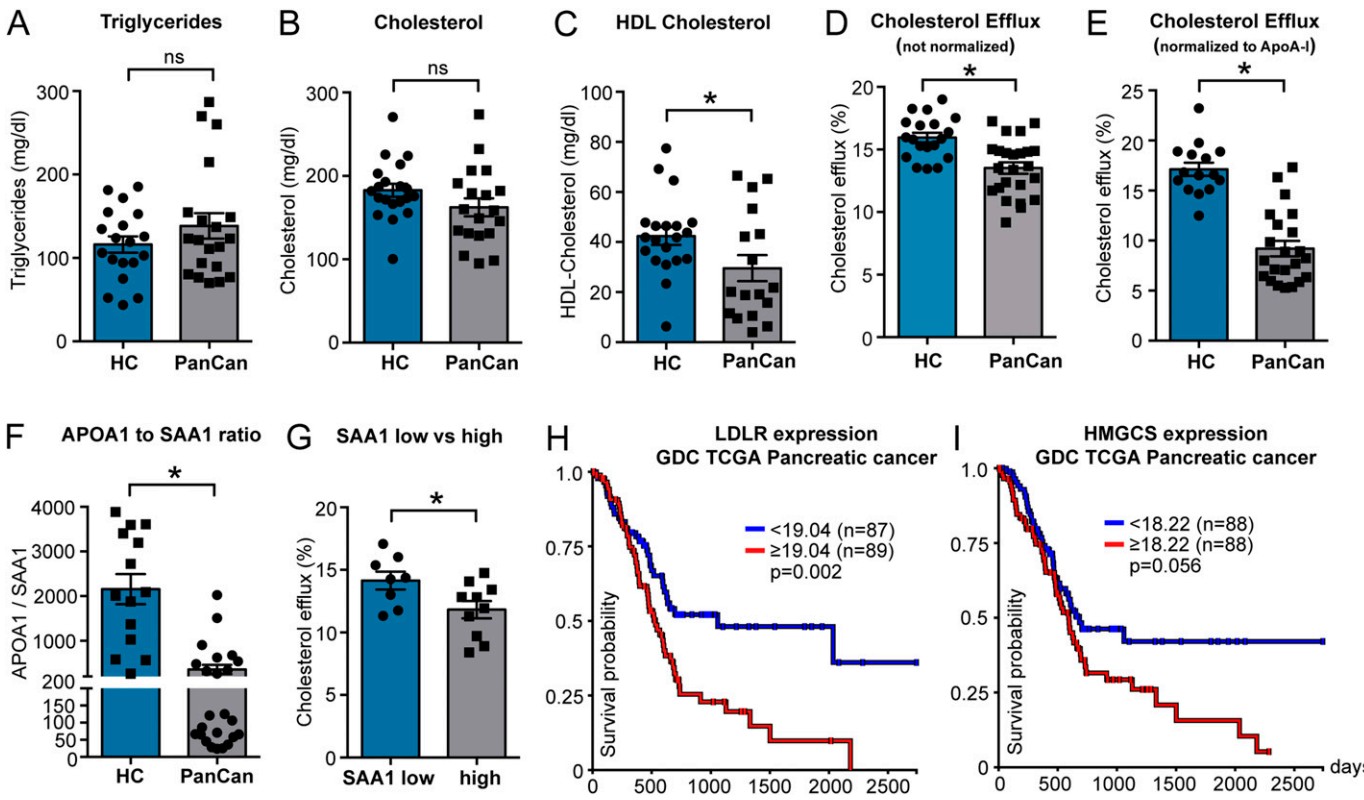

**Figure 6. Pancreatic-cancer patients show decreased plasma high-density lipoprotein-C and cholesterol efflux capacities compared with healthy donors and increased survival probability with lower tumor-associated low-density lipoprotein receptor and HMGCS expression levels.**
**(A, B, C)** Plasma total triglyceride levels, (B) total cholesterol levels, and (C) high-density lipoprotein-C were determined in plasma samples from a cohort of healthy volunteers and pancreatic-cancer patients in late stages of their disease (n = 19/19; *P < 0.05; unpaired *t* test). **(D, E)** Nonnormalized plasma efflux capacity and (E) plasma efflux capacity normalized to APOA1 levels (n = 19/19; *P < 0.05; unpaired *t* test). **(F)** APOA1-to-SAA1 plasma protein ratio after ELISA measurement of protein concentrations. **(G)** Cholesterol efflux in plasma samples of patients with low (<13 μg/ml) or high (>13 μg/ml) SAA1 plasma concentrations. **(H, I)** Kaplan–Meier overall survival plots for low-density lipoprotein receptor and HMGCS expression, respectively, in the GDA TCGA pancreatic-cancer cohort (log-rank test, *P < 0.05, UCSC Xena).

inverse correlation of the *LDLR* and *HMGCS* with overall survival persisted with high significance when analyzing the TCGA Pancancer database (Fig S4C and D). Together, the decreased efflux capacity of pancreatic-cancer patient plasma samples and the inverse association of patient overall survival with genes that promote intracellular cholesterol accumulation indicate the activation of metabolic mechanisms that favor the availability of cholesterol for cancer cells, which might eventually support PDAC malignancy.

# Discussion

The here presented data provide evidence that efficient cellular cholesterol removal mediated by discoidal, lipid-poor reconstituted HDL particles reduces pancreatic-cancer cell growth and might hold the potential to attenuate the development and spread of the disease. Although SR-B1 is highly expressed in the applied pancreatic-cancer cell lines and significantly contributes to HDL-mediated cholesterol flux, forced efflux via ABCA1 increases the antitumor activity of those particles (see graphical abstract). In

contrast, lipid-laden, spherical-like HDL particles, which exhibit a higher affinity for SR-B1-directed lipid exchange at the plasma membrane, showed reduced or insignificant ability to counteract cancer cell proliferation and tumor growth (see graphical abstract). Unidirectional, partially ABCA1-driven cholesterol efflux via disc-shaped rHDL particles might thereby cause depletion of cellular cholesterol pools, eventually leading to decreased proliferation and viability of cancer cells. These results indicate that the HDL particle composition and thereby HDL functional metrics might determine its antitumor capacity.

In the field of cardiovascular research, HDL functionality is currently under intense investigation, as the gold-standard plasma parameter, the level of HDL-C has been shown not to correlate with cardiovascular risk in interventional trials (36). One potential explanation for this discrepancy is the vast heterogeneity of the HDL particle pool. HDLs appear in the plasma as small, lipid-poor pre-β HDL and lipid-enriched α-HDL particles. Thereby, the pre-β fraction, which only comprises about 5% of total HDLs in the circulation, performs net cholesterol efflux from peripheral cells, predominantly macrophage foam cells, to eventually become α-HDLs (9). This α-HDL fraction is enriched in phospholipids, CE, and triglycerides and acts as a high-affinity ligand for SR-B1, which, under

physiological conditions, serves as a receptor on hepatocytes that binds and sequesters HDL-associated cholesterol for excretion (21). SR-B1 in turn was previously shown to be overexpressed in many cancer entities including pancreatic cancer (37, 38). Therefore, as the results from this study indicate, spherical and lipid-rich HDL particles might serve as cholesterol and CE source for cancer cells, eventually used as cellular fuel to drive cancer cell proliferation. To this end, patient lipid and lipoprotein profiles might significantly contribute to cancer progression. Interestingly, obesity and the metabolic syndrome are well recognized risk factors for the development of tumor malignancies of various types (39).

Efforts to artificially increase the cholesterol-removing pre-$\beta$/discoidal HDL fraction to drive ABCA1-mediated net cholesterol depletion from cancer cells might therefore provide a molecular axis that offers therapeutic potential for the treatment of PDAC. In support of this hypothesis, LXR agonists, which are potent activators of ABCA1 expression and thereby cholesterol efflux, reduced proliferation, cell cycle progression, and colony formation of human PDAC cell lines (40). Moreover, LXR agonists have also been demonstrated to increase the expression of ABCA1 and the induction of the LDLR-degrading ubiquitin E3 ligase IDOL, which leads to tumor cell apoptosis and a reduction in tumor growth in glioblastoma xenograft models (41). Of note, high levels of *LDLR* and low levels of *IDOL* (*MYLIP*) expression correlate with a worse prognosis in patients suffering from pancreatic cancer and other tumor entities (Figs 6G and S4C and D). Cholesterol depletion by the use of statins was also shown to inhibit gallbladder cancer cell proliferation and sensitized those cells to cisplatin treatment, possibly by the inhibition of the DNA repair machinery (42). In view of those data, experiments which combine the administration of LXR agonists and efficient cholesterol acceptors such as rHDL1-like particles with standard of care chemotherapy will provide valuable insights concerning the therapeutic applicability of the here presented preclinical findings.

Native HDL particles possess anti-apoptotic capacity which has been shown for cell types such as macrophages and endothelial cells (43, 44, 45). Accordingly, in most of the performed experiments in this study, the pro-apoptotic effect of native HDL is small or even absent. Our data rather suggest that reconstituted, discoidal HDLs have a more pronounced capability to induce apoptosis in pancreatic-cancer cells. One possible explanation for the apoptosis-inducing effect of rHDL1 particles might be the increased sensitivity of certain cancer cell lines to cholesterol depletion and disintegration of lipid rafts, specific plasma membrane microdomains required for the efficient concentration, and activation of pro-proliferative signaling cascades such as the Ras/ERK/MAPK or AKT pathways (reviewed in references 46 and 47). A broadly observed phenomenon in cancer cells is the constitutive activation of AKT signaling, resulting in BAD phosphorylation, thereby deactivating its pro-apoptotic function (reviewed in reference 48). In turn, the observed reduction in AKT activation (Fig 5H) might lead to BAD-induced apoptosis of rHDL1-treated pancreatic-cancer cells.

As mentioned in the introduction, clinical studies indicate an inverse association of HDL-C with cancer incidence of multiple entities. Interestingly, data presented here point toward a decrease in the cholesterol efflux capacity of plasma samples from cancer patients (Fig 6D and F). In addition to a reduction in HDL quantity,

tumors might also be capable of influencing HDL functionality. HDL-associated proteins such paraoxonase 1 (PON1) and serum amyloid A (SAA) and biochemical modifications of HDL structural components such as myeloperoxidase (MPO)-mediated nitration or chlorination of APOA1 are currently known to influence HDL's reverse cholesterol transport capacity. The overexpression of PON1, an HDL-associated enzyme with potent anti-oxidative activity, has been shown to increase HDL-C efflux in vitro and reverse cholesterol transport in vivo (49). Of note, PON1 serum activity is reduced in cancer patients of various entities (37). SAA1 is an acute phase protein and transported predominantly on HDL in the bloodstream. Upon infection, SAA1 levels increase dramatically and its association with HDL has been demonstrated to reduce the lipoproteins' anti-inflammatory properties and its cholesterol efflux capacity (34, 37). Here, we observed high SAA1 levels and a reduction of the APOA1-to-SAA1 plasma ratio in patients with pancreatic cancer (Fig 6). When stratifying within the cancer patient cohort, cholesterol efflux capacity was significantly decreased in plasma samples with high SAA1 levels, indicating a physiologic relevance of SAA1-mediated HDL remodeling in PDAC. Interestingly, certain cancer cell lines, tumor-associated macrophages, and pancreatic cancer–associated adipocytes produce large amounts of SAA (37, 50). SAA levels were furthermore shown to directly correlate with disease progression, reduced survival rate, and poor overall prognosis (37). In addition, macrophages and myeloid-derived suppressor cells accumulating in cancer patients express high levels of the enzyme MPO, a candidate enzyme that oxidatively modifies HDL, thereby reducing its cholesterol efflux capacity (36, 51). Interestingly, MPO-mediated HDL modifications enhanced association of HDL with macrophages in cell culture and increased cholesteryl ether transfer into target cells in an SR-B1–dependent manner (51). If this scenario is likely to happen in the tumor microenvironment, oxidatively modified HDL particles, although losing their cholesterol removal capacity, might serve as efficient cholesterol donors in SR-B1–expressing cancer entities such as pancreatic cancer.

Although an antitumor effect for AAV-delivered APOA1 and therapeutically administered rHDL particles could be demonstrated, there are limitations to the here presented in vivo experiments. Regarding AAV-delivered APOA1/HDL, it was not possible to analyze those particles regarding their structural and/or biochemical composition. Interestingly, a study by Lebherz and Rader showed that the hepatic expression of full-length human APOA1 and the APOA1 Milano variant using AAV2.8 particles reduced HDL cholesterol levels in WT mice while reducing atherosclerotic burden (52). Although the detailed mechanism of this observation remained ambiguous, the authors hypothesize that HDL particle catabolism and therefore HDL-mediated cholesterol efflux might increase (52). One hint that HDL catabolism might be increased in tumor-bearing APOA1 KO mice reconstituted with AAV-APOA1 particles is reflected by low HDL cholesterol levels compared with WT animals albeit the robust expression of APOA1 in those animals (Fig 5C of the revised manuscript). To definitely proof this hypothesis, HDL turnover/in vivo reverse cholesterol transport studies (53) with $^3$H cholesterol labeled tumor cells would be required to analyze the potential increase in AAV-delivered HDL catabolism. The rather small antitumor effect of rHDL particles and

AAV-delivered HDL particles in WT mice could be explained by (i) the highly aggressive growth characteristics of Panc02 cells, (ii) fast initiation of tumor growth paralleled by a late onset of AAV-transgene expression/low amounts of injected rHDL particles (compared with amounts used in, e.g., reference 3), and (iii) poor vascularized tumors, which hampers the accessibility of AAV-delivered HDL particles to tumor cells.

In summary, the presented data demonstrate a potentially important role of HDL-mediated cholesterol efflux in reducing the proliferative capacity of PDAC cells. Thereby, the HDL particle composition might dictate its antitumor activity by regulating directionality of net cholesterol flow between cancer cells and the lipoprotein particle.

# Materials and Methods

### Animals

C57Bl6/J *Apoa1* KO mice were from the Jackson Laboratory (B6.129P2-*Apoa1tm1Unc*/J). *Apoa1* KO mice were bred heterozygously, and the wild-type littermates were used as controls. All animal experiments were carried out in concordance with the institutional guidelines for the welfare of animals and were approved by the local licensing authority Hamburg (project number G36/13 and G126/15). Housing, breeding, and experiments were performed with animals between 10 and 16 wk of age under a 12-h light – 12-h dark cycle and standard laboratory conditions (22°C ± 1°C, 55% humidity, food, and water ad libitum).

### Cell lines and culture conditions

The murine pancreatic adenocarcinoma cell lines Panc02 and 6066 were a kind gift of Dr. Lars Ivo Partecke (Schleswig) and were maintained in RPMI medium supplemented with 10% FCS, 2 mM L-glutamine, 100 U/ml penicillin, and 100 $\mu$g/ml streptomycin (complete RPMI). The human pancreatic adenocarcinoma cell line BxPC3 was from ATCC and maintained in complete RPMI medium. The murine melanoma cell line B16F10, Lewis LLCs, and the breast adenocarcinoma cell line E0771 were a kind gift of Prof. Dr. Peter Carmeliet and Prof. Massimiliano Mazzone (VIB Vesalius Research Center, KU Leuven) and maintained in DMEM medium supplemented with 10% FCS, 2 mM L-glutamine, 100 U/ml penicillin, and 100 $\mu$g/ml streptomycin (complete DMEM). Cells were maintained at 37°C and 5% $CO_2$ in a humidified atmosphere and routinely tested to be mycoplasma negative (MycoAlert Mycoplasma Detection Kit; Lonza). Cells were cultured no longer than 15 passages before experimental use.

### Cholesterol depletion assays

Panc02, 6066, and BxPC3 cells (1 × 10⁴ cells per well of a 96 well plate) were seeded in RPMI medium containing either 2% FCS, 2% lipoprotein-deficient FCS (LPDS), or 2% LPDS containing 5 $\mu$M lovastatin and 100 $\mu$M mevalonate for cholesterol depletion (54, 55). Cells were grown for 96 h, and cellular viability was determined at

indicated time points using the cell proliferation reagent WST1 (Roche) according to the manufacturer's instructions.

### Preparation of reconstituted HDL (rHDL) particles

Native human HDL was isolated from healthy donors using serial density ultracentrifugation as described (56, 57). Reconstituted HDL particles were prepared according to the method of Jonas et al (58, 59). Briefly, 1 mg of native HDL was delipidated twice using 5 ml ethanol:diethyl ether (3:2). The supernatant was discarded, and the remaining solvents were evaporated with nitrogen gas. Phosphatidylcholine (PC), cholesterol (C), and cholesteryl palmitate (CE, in chloroform:methanol (2:1)) were combined in specific molar ratios, and the solvents were evaporated with nitrogen gas. The dried lipids were resuspended in 200 $\mu$l buffer A (150 mM NaCl, 0.01% EDTA, 10 mM Tris–HCl, pH 8.0), and 50 $\mu$l of a 30 mg/ml sodium deoxycholate solution was added to disperse the lipids. The mixture was stirred at 4°C for 2 h. Delipidated HDL was dissolved in 250 $\mu$l of buffer A. Both suspensions were mixed in a glass vial and stirred at 4°C overnight. The suspension was then filtered twice through a 4-ml 3K Amicon filter tube, and the protein concentration was determined. Reconstituted HDL was overlaid with nitrogen gas and stored at 4°C. To prepare tracer-labeled rHDL, 25 $\mu$Ci ³H cholesteryl oleate (NET746L001MC; Perkin Elmer) and 12.5 $\mu$Ci ¹⁴C cholesterol (CFA128; Amersham) dissolved in toluene were added to the lipid mixture (for an equivalent of delipidated APOA1 of 500 $\mu$g) before evaporation. Total cholesterol and phospholipid content of native and rHDL particles were determined with kits from DiaSys. To confirm particle homogeneity, HDL particles were separated on 0.5% native Agarose gel run in TAE buffer pH8.6. After fixation in ethanol:acetic acid:$H_2O$ = 60:10:30 for 1 h at room temperature, the gel was stained with Coomassie blue. Particles were subjected to 12% reducing SDS–PAA gel electrophoresis and subsequent Coomassie blue staining for the analysis of protein composition.

### AFM imaging and particle analysis

AFM measurements were performed using an atomic force microscope (JPK BioAFM-NaonWizard 4; JPK). Silicon nitride AFM probes with a nominal spring constant of 0.6 N/m and a nominal tip radius of 20–60 nm (MLCT-F; Bruker Nano Inc.) were used for the measurements. The exact sensitivity and spring constant of each cantilever were determined on a cleaned coverslip in 200 $\mu$l PBS using a force-displacement experiment and a thermal noise spectrum measurement. All samples were diluted at a ratio of 1:1,000. A volume of 200 $\mu$l of the diluted HDL solution was incubated on the cleaned glass coverslip for at least 15 min and then imaged. AFM images were acquired using advanced imaging software (quantitative imaging mode—QI-mode) from Bruker. A maximum set point force of <1 nN was used. Particle, half-width (FWHM), and probe molecule height analysis was performed using JPK Data Processing software (version 6.1.163; JPK). Convolutions of tip artifacts were corrected (60). Height and width were fitted using a Gaussian mixed distribution model to determine the underlying distributions and their associated height and width. The AR (in percent) was calculated as previously described (60). An AR of 100%

represents a perfect spherical shape; lower values indicate prone discs.

## (r)HDL treatments of pancreatic adenocarcinoma cell lines

Panc02 and 6066 cells were starved in a T75 flask overnight in RPMI medium containing 0.1% FCS. Next, $1 \times 10^4$ cells were seeded per well of a 96-well plate in RPMI medium containing 2% FCS with the addition of indicated HDL particles (75 µg/ml). To inhibit SR-B1, BLT1 (0.5 µM) was added 30 min before the addition of HDL to the cell suspension. For ABCA1 activation, TO901317 (or DMSO control) was added at the indicated concentrations directly to the starvation medium the day before the assay and throughout the experiment.

## Gas chromatography

Cells ($0.8 \times 10^6$) were seeded in 6-cm dishes in RPMI medium supplemented with 2% FCS, 1% Pen/Strep, and 1% glutamine and grown overnight. Afterward, cells were treated with HDL particles (75 µg/ml) in the presence of 5 µM lovastatin for 48 h. Afterward, cells were washed, detached by trypsin/EDTA, resuspended in PBS, and centrifuged (4°C, 200g, 5 min). Lipids were isolated from cell pellets by standard Folch extraction. An aliquot of the pellet was lysed with NaOH (0.1 mol/l) and used for cell protein determination by the Bradford assay. Lipids were separated by gas chromatography as described previously (61). Tridecanoyl glycerol, cholesteryl myristate, and trinonadecanoyl glycerol (all from Sigma-Aldrich) were used as standards for free cholesterol, esterified cholesterol, and triglycerides, respectively. Values were normalized to cell protein.

## Quantitative PCR

Total RNA was isolated from cells or murine liver tissue using the Relia Prep RNA Tissue Miniprep System (Promega) according to the manufacturer's instructions. Then 1 µg of RNA was reverse-transcribed into single-stranded cDNA (GoScript Reverse Transcription System; Promega) and subsequently used for qPCR analyses on a StepOnePlus Real-Time PCR Detection System (Applied Biosystems). Expression levels of genes of interest were normalized to hypoxanthine guanine phosphoribosyl transferase (*Hprt*), and the relative fold gene expression compared with control was calculated using the $2^{-(\Delta\Delta Ct)}$ method.

## Western blotting

A total of 0.5 µl of mouse plasma or 20 µg of Panc02 RIPA total protein extracts were analyzed by reducing SDS–PAGE (8% PAA gel) and transferred to nitrocellulose membranes. Nonspecific binding sites were blocked with TBS (20 mM Tris–HCl, pH 7.4, 137 mM NaCl) containing 5% (wt/vol) fatty acid–free BSA or nonfat dry milk and 0.1% Tween-20 (blocking buffer) for 1 h at room temperature. Proteins of interest were detected with antibodies directed against APOA1 (in-house produced rabbit polyclonal antihuman APOA1 antibody), ABCA1 (MAB10005; Merck), β-actin (clone AC-74; Sigma-Aldrich), phospho-AKT (S437, 4060S; Cell Signaling), phospho-AKT (Thr308, 13038S; Cell Signaling), and pan AKT (2920S; Cell Signaling) followed by incubation with HRP-conjugated secondary antibodies

and development with the enhanced chemiluminescence protocol (Pierce).

## Cholesterol flux assays

To measure cholesterol efflux from Panc02 or 6066 cells to HDL particles, $0.1 \times 10^6$ cells were seeded in 900 µl of complete RPMI per well of a 12-well plate and incubated for 24 h at 37°C and 5% $CO_2$. Next, cells were trace-labeled for another 24 h with $^3H$ cholesterol (NET139; Perkin Elmer) by adding 100 µl of complete RPMI containing 5 µCi $^3H$ cholesterol/ml per well. The next day, cells were washed twice with warm RPMI medium containing 0.1% FCS and once with 1 ml of warm PBS. After carefully removing the medium, 500 µl of 0.1% FCS-containing RPMI containing HDL particles (10 µg/ml) was added to the cells. When analyzing the cholesterol acceptor capacity of human plasma samples, instead of HDL particles, 10 µl (2%) of plasma was added to 500 µl of 0.1% FCS-containing RPMI. In SR-B1 inhibition studies, BLT1 (1 µM) was added 1 h before the addition of efflux acceptors. For the activation of ABCA1 expression, the LXR agonist TO901317 (5 µM) was added to cells 48 h before the addition of efflux acceptors. Efflux acceptors were incubated for 8 h with the cells. To analyze the transfer of cholesterol and its esters from HDL to the cellular compartment, tracer-labeled HDL particles (10 µg/ml) were again diluted in 0.1% FCS-containing RPMI medium and incubated for 8 h with the cells. Supernatants are collected and cleared from cellular debris by centrifugation at 10,000g for 10 min at room temperature. Cells were washed twice with PBS and lysed by the addition of 500 µl 0.1 N NaOH. A total of 200 µl of either supernatant or cell lysate were mixed with 8 ml of Ultima Gold scintillation cocktail and analyzed by scintillation counting.

## AAV particle production

The production of liver-targeting AAV particles of the serotype AAV2.8 was performed as previously described in detail (62, 63, 64). Briefly, the full-length murine APOA1 cDNA was inserted into a pAAV-MCS plasmid containing AAV inverted terminal repeats using BstBI (fwd) and BsrGI (rev) restriction enzyme sites. Together with a pAAV rep2 cap8 transfer plasmid and an AdpXX6 helper plasmid, HEK cells were co-transfected, and virus particles purified from cell pellets and supernatants using iodixanol density gradients (62, 64).

## In vivo experiments

To compare tumor growth kinetics in WT and *Apoa1* KO mice, $0.5 \times 10^6$ Panc02, B16F10, or LLC cells were implanted subcutaneously into the right flank of mice. E0771 cells were implanted orthotopically into the second mammary fat pad. Tumor size was measured with a digital caliper, and the volume was calculated using the formula V = (length$^2$ × width)/2. For histological analyses, pimonidazole (1 mg, i.p.) was injected 2 h before euthanasia. For AAV-mediated reconstitution of hepatic APOA1 levels in *Apoa1* KO mice, $2 \times 10^{11}$ AAV2.8 particles encoding the full-length murine APOA1 mRNA (AAV-APOA1) were administered intravenously 5 d before tumor cell inoculation. For rHDL injection studies, tumors were grown to a size of 100 mm$^3$. Afterward, mice received intravenous injections of either 0.2 mg rHDL (PC:C:CE:APOA1 = 100:12.5:0:1; ZLB Behring, a kind

gift of Prof. Matti Jauhiainen) or an equivalent volume of sterile PBS every 72 h. Alternatively, $2 \times 10^6$ Panc02 cells were administrated intraperitoneally into female C57Bl6/J mice. After 5 d, rHDL1 (50 mg/kg) or PBS control was administered every other day by intraperitoneal injections.

## Collection of plasma samples and analysis of lipid parameters

Human blood samples were collected into precoated EDTA tubes under informed consent of patients and approval by and strict adherence to institutional guidelines of the Medical University of Vienna and the Declaration of Helsinki (Ethik Votum 1035/2020). The cohort of pancreatic-cancer samples included 20 patients of mixed sex in advanced stages of their disease (10 female, 10 male) with a mean age of 69 and a mean BMI of 24.6. Murine blood samples were obtained by retroorbital bleeding and collection of blood into precoated EDTA tubes (Sarstedt). Blood samples were immediately centrifuged for 15 min at 3,000$g$ at room temperature, and plasma samples were collected and frozen at –80°C until further use. Total plasma triglycerides, cholesterol, and HDL-C were measured with the Triglyceride FS, Cholesterol FS, and HDL-C Immuno FS kit systems (DiaSys). APOA1 and SAA1 plasma levels were determined with DuoSet ELISA Kits (R and D Systems) according to the manufacturer's instructions. Plasma samples were diluted 1:10,000 for APOA1 and 1:100 for SAA1 before analysis.

## Analysis of intratumoral hypoxia

Tumor samples were fixed overnight in 4% paraformaldehyde at 4°C and embedded in paraffin. Paraffin sections (4 $\mu$m) were stained with antibodies to detect tumor hypoxia (pimonidazole, HP3-1000kit) as previously described (65). For morphometric analysis, 8–10 optical fields per tumor section were acquired using a Zeiss Axio Scope A1, and images were analyzed using the NIH ImageJ software.

## Flow cytometry

Flow cytometric analysis of enzymatically digested tumor tissue was essentially performed as described in reference 65. MRC1+ macrophages were gated as PE-Cy7 CD11b[+] (clone M1/70 BD Bioscience), PE F4/80[+] (clone BM8; BioLegend), and FITC CD206 (MRC1)[+] (clone C068C2; BioLegend). Granulocytic myeloid-derived suppressor cells (GMDSC) were gated as PE-Cy7 CD11b[+], PE Ly6-G[+] (clone 1A8; BD Bioscience), and PerCP-Cy5.5 Ly6-C[int] (clone Hk1.4; eBioscience). T cells were characterized as APC CD3[+] (clone 17A2; eBioscience) and either eFluor 450 CD8a[−] (clone 53-6.7; eBioscience) FITC CD4[+] (clone GK1.5; BioLegend) for T-helper cells or vice versa for cytotoxic T cells. DAPI was used as a viability stain.

## Detection of cellular apoptosis

Trypsinized and washed Panc02 cells were stained with the FITC Annexin V Apoptosis Detection Kit with 7-AAD (BioLegend) and analyzed using the CytoFLEX S Flow Cytometer (Beckman Coulter). Alternatively, Panc02 cells were serum starved for 24 h and afterward plated in 96-well plates ($1 \times 10^4$ cells/well) in the presence of 2% FCS and treated with different HDL particles in the indicated concentrations. Then 24 h afterward, cellular apoptosis was measured with the Caspase-Glo 3/7 assay (Promega) according to the manufacturer's instructions.

## Statistics

To compare the means of two groups, an unpaired, two-tailed $t$ test was used. Pairwise comparison testing in experiments with more than two groups was performed using one-way ANOVA followed by the Tukey post hoc test for multiple comparisons. Pairwise comparisons of tumor growth kinetics were performed using two-way ANOVA followed by the Bonferroni post hoc test for multiple comparisons. Normality testing was performed using the Shapiro–Wilk test. All statistical calculations were performed using GraphPad Prism 9, and data represent mean ± SEM of representative experiments. All experiments (except mouse studies) were repeated 2–3 times; indicated n numbers represent biological replicates. In all cases, statistical significance was assumed when $P < 0.05$.

# Data Availability

All presented, described, and discussed data are contained within this manuscript. Source data and original images are available upon request.

# Supplementary Information

# Acknowledgments

The authors would like to thank Agnes Hunger, Franziska Pupp, Victoria Guggenberger, and Anna-Lena Höbler for excellent technical assistance and Ernst Steyrer and Wolfgang Sattler (both Medical University of Graz) for support and fruitful discussions in the process of conceptualizing and preparing this manuscript. R Oberle was supported by an Erwin-Schrödinger fellowship of the Austrian Science Fund (FWF, J3664-B19). F Udonta received a Werner Otto fellowship from the Werner Otto foundation. M Wroblewski was supported by the medical faculty of the University of Hamburg (FFM program). S Loges is supported by the European Research Council (ERC) under the European Union's Horizon 2020 research and innovation programme (Grant Agreement No. 758713) and by the Hector Foundation II. B Plochberger and F Weber are supported by the Austrian Science Fund (FWF, P33481-B).

## Author Contributions

R Oberle: conceptualization, data curation, supervision, funding acquisition, investigation, visualization, methodology, project administration, and writing—original draft, review, and editing.
K Kuhrer: data curation, investigation, and methodology.
T Osterreicher: data curation and methodology.
F Weber: data curation and methodology.

S Steinbauer: data curation and methodology.

F Udonta: data curation and methodology.

M Wroblewski: conceptualization and methodology.

I Ben-Batalla: data curation and methodology.

I Hassl: data curation and methodology.

J Korbelin: conceptualization and methodology.

M Unseld: resources and data curation.

M Jauhiainen: conceptualization and methodology.

B Plochberger: resources and methodology.

C Rohrl: conceptualization, resources, and methodology.

M Hengstschlager: resources and formal analysis.

S Loges: conceptualization, resources, and methodology.

H Stangl: conceptualization, resources, formal analysis, and methodology.

## Conflict of Interest Statement

The authors declare that they have no conflict of interest.

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
