## [Reviewer comments · Life Science Alliance]

Life Science Alliance

The HDL particle composition determines its anti-tumor activity in pancreatic cancer

Raimund Oberle, Kristina Kühner, Tamina Österreicher, Florian Weber, Stefanie Steinbauer, Florian Udonta, Mark Wroblewski, Isabel Ben-Batalla, Ingrid Hassl, Jakob Körbelin, Matthias Unseld, Matti Jauhiainen, Birgit Plochberger, Clemens Röhrl, Markus Hengstschläger, Sonja Loges, and Herbert Stangl

DOI: <https://doi.org/10.26508/lsa.202101317>

Corresponding author(s): Raimund Oberle, Medical University of Vienna

Review Timeline:

Submission Date:	2021-11-25
Editorial Decision:	2021-12-20
Revision Received:	2022-04-04
Editorial Decision:	2022-04-25
Revision Received:	2022-05-02
Accepted:	2022-05-03

Transaction Report:

December 20, 2021

Re: Life Science Alliance manuscript #LSA-2021-01317-T

Dr. Raimund Oberle
Medical University of Vienna
Institute of Medical Chemistry, Center for Pathobiochemistry and Genetics
Währingerstraße 10
Vienna, Vienna 1090
Austria

Dear Dr. Oberle,

Thank you for submitting your manuscript entitled "The HDL particle composition determines its anti-tumor activity in pancreatic cancer" to Life Science Alliance. The manuscript was assessed by expert reviewers, whose comments are appended to this letter. We invite you to submit a revised manuscript addressing the Reviewer comments.

Thank you for this interesting contribution to Life Science Alliance. We are looking forward to receiving your revised manuscript.

Sincerely,

B. MANUSCRIPT ORGANIZATION AND FORMATTING:

Reviewer #1 (Comments to the Authors (Required)):

The authors did a comprehensive analysis of HDL interactions with prostate cancer cell lines and explored the relevance of their findings in a clinical cohort. The data are interesting but limited by several shortcomings in the evaluation and interpretation:

1. page 1, lines 150 ff.: The authors reconstituted artificial HDL, one poor in cholesterol and one enriched in cholesterol. They claim that the latter is spherical but do not show any proving data. Figure 1D is a cartoon only. For rHDL2 the authors included cholesteryl esters that are needed to form spherical esters. However, whether the simple co-incubation was sufficient to produce a cholesteryl ester core is not shown and questionable. Usually purified CETP or a source of CETP such as lipoprotein free plasma as well as a lipoprotein donor (usually LDL) are used to reach this. The authors only show the higher cholesterol content of rHDL2 compared to rHDL1. However, this may be unesterified cholesterol only. Then the resulting particle will be discoidal rather than spherical. Negative stain electronmicroscopy is needed to prove the different shapes of rHDL1 and rHDL2.
2. (page 8) The authors show that HDL promotes apoptosis of the tumor cells. This is a very atypical effect of HDL. HDL is rather known to inhibit apoptosis. This has been shown in multiple cell types. Thus the proapoptotic effects need to be substantiated, ideally by showing the mechanism to make this different and unexpected behaviour plausible. In addition, the authors should discuss this discrepancy
3. (page 9). The authors used double labeled lipoproteins to estimate the flux of cholesterol and cholesteryl esters. Ideally, the authors determine net cholesterol fluxes by measuring cholesterol mass concentrations in both cells and media. This will answer the important Question on the relative contribution of cholesterol influx and efflux by HDL
4. page 11: the in vivo experiment is important and interesting. Since wild type mice and apoA1 ko mice do not differ towards tumor growth but overexpression of APOA1 or injection of rHDL reduces tumor growth in apoA1 ko mice it will be important to show the effect of APOA1 expression and rHDL in wild type mice as well. If the authors hypothesis of the need of distinct HDL particles is right, also this Intervention will reduce tumor growth. An alternative explanation is that human apoA-I but not murine apoA-I₂ protects. This should be tested by according experiments in vitro
5. (page 12, lines 281ff) the authors measured cholesterol efflux capacity. They only present data on cholesterol efflux capacity normalized to apoA-I levels. They must show the non-normalized data at least as well, if not only. They are more important because this reflects the physiological or pathological situation of Plasma. Moreover, by normalization, differences can be due to differences in cholesterol efflux capacity, apoA-I levels or both. The answer will determine the interpretation
6. (page 13). the Kaplan Meier analysis is interesting. However, in the context of the present study, it will be most interesting to see if SCARB1 and ABCA1 expression is associated with survival. If not, the interpretation will have to be substantially changed.
7. page 14 lines 324 ff and page 15 line 1ff. The authors discuss prebeta1 HDL. Thereby they claim that rHDL1 is an analog of prebeta1 HDL. This is principally wrong, because prebeta1 HDL corresponds to lipid-free or poor apoA-I. rHDL1 contains 100 moles of PC per mole of apoA-I and has alpha rather than prebeta mobility as also shown by the authors in Figure 1G. Looking to this figure one sees that rHDL1 migrates faster rather than slower than native HDL and is a prealpha- rather than a prebeta-particle. In fact, rHDL1 is an analogue of the ABCA1 product (rather than educt) and the LCAT educt.
8. page 15, lines 349 to 353: as discussed in 6., the authors must show non-normalized data to make this claim
9. page 18, see my comment 1 on rHDL2 and cholesteryl ester incorporation
10. page 22: The authors use parametric statistical Tests without showing normality. The low numbers of in vitro experiments make non-parametric tests preferable anyway. More and more journals request this, for example if $N < 6$.

Comment to Review 2:

This review is more positive than mine. It appears to disagree in some aspects with mine and also makes some interesting suggestions:

1. I like the Suggestion on figure 1 to use Cyclodextrins as a comparator
 2. By contrast to Reviewer 2, I still have issues with the characterization of rHDL2 (spherical shape and successful incorporation of cholesteryl esters must be shown) and the interpretation of rHDL1 as an analogue of prebeta1-HDL. The authors disprove this frequent misunderstanding by themselves with figure 1G
- figure 4 will benefit from cholesterol mass determination
figure 5: I agree with the concerns of reviewer 2 and made suggestions to address them
Figure 6: agree with the need of providing patient characteristics and the need of defining % cholesterol efflux. And as stated by me before, please provide data on non-adjusted cholesterol efflux capacity. Finally, I consider data on survival stratified by ABCA1 and SCARB1 expression as essential

Reviewer #2 (Comments to the Authors (Required)):

The authors document anti-tumor potential of HDL-mediated cholesterol efflux and use elegant synthetic biochemistry to dissect HDL heterogeneity and identify lipid composition as a major factor involved in cholesterol unidirectional trafficking and anti-tumor activity.

The study is well presented and neatly organized. Experiments are well performed and controls are in place. I think this is an excellent work that significantly add to our understanding of the complex role of cholesterol dynamics in cancer. I only have minor (some not essential) comments:

- Figure 1 (first part) makes the case that cholesterol depletion inhibits tumor growth in vitro. However, conditions for cholesterol depletion in Fig 1A are not clear. Why authors add mevalonate (which is a cholesterol precursor and can potentially support endogenous biosynthesis)? Could they do that with cyclodextrins?
- Figure 1 (second part): description of rHDL composition; no issues
- Figure 2: HDL composition affects viability. Excellent experimentation. No major issues, although I would have preferred analysis of SREB activation rather qPCR of downstream targets. But data look OK. Please, clarify that data in F thru H express fold change over CONTROL.
- Figure 4 explains the HDL composition dictates cholesterol flux direction and impacts tumor cell viability. No issues, authors are commended for clever strategy. However, definitive proof for ABCA1 role would require knockdown experiments, so maybe authors want to comment that.
- Figure 5 describe therapeutic potential of rHDL. Results are neat and experiments well performed but I don't understand the observed differences between WT, ABCA1 KO and AAV-reconstituted mice. APO levels appear similar between WT and AAV, so why the difference in tumor growth? Authors do state native HDL appear irrelevant for tumor growth, so where's the change? AAV-transduce mice produce different HDL? Shouldn't the authors show data for that? Is that somehow already known? I think the authors should either comment a bit more on that or show that HDL particles are indeed different in the two cohorts.
- Figure 6 shows cholesterol parameters in PDAC patients. I couldn't find description of the patients cohort anywhere, nor compliance with ethical requirements. Please add that. Not clear how they calculate cholesterol efflux % in D.
- Figure 7: model

"The HDL particle composition determines its anti-tumor activity in pancreatic cancer"**submitted by**

Raimund Oberle, Kristina Kühner, Tamina Österreicher, Florian Weber, Stefanie Steinbauer, Florian Udonta, Mark Wroblewski, Isabel Ben-Batalla, Ingrid Hassl, Jakob Körbelin, Matthias Unseld, Matti Jauhiainen, Birgit Plochberger, Clemens Röhr, Markus Hengstschläger, Sonja Loges, Herbert Stangl

General Remark: The authors would like to thank the reviewers for the effort they put in reviewing this manuscript and for their insightful comments. As requested, the author addressed all concerns and critiques of the referees in a point-by-point response. The authors are enthusiastic about the improvements of the manuscript and believe that the scientific quality could be significantly increased.

REVIEWER A:

1. *page 1, lines 150 ff.: The authors reconstituted artificial HDL, one poor in cholesterol and one enriched in cholesterol. They claim that the latter is spherical but do not show any proving data. Figure 1D is a cartoon only. For rHDL2 the authors included cholesteryl esters that are needed to form spherical esters. However, whether the simple co-incubation was sufficient to produce a cholesteryl ester core is not shown and questionable. Usually purified CETP or a source of CETP such as lipoprotein free plasma as well as a lipoprotein donor (usually LDL) are used to reach this. The authors only show the higher cholesterol content of rHDL2 compared to rHDL1. However, this may be unesterified cholesterol only. Then the resulting particle will be discoidal rather than spherical. Negative stain electronmicroscopy is needed to prove the different shapes of rHDL1 and rHDL2.*

Response: The authors agree with the concerns of the reviewer that the analysis of the artificially produced rHDL particles was insufficient. To improve this important aspect, we performed analysis of the particles using atomic force microscopy (AFM) to determine particle shape and size. Moreover, we measured cholesterol and cholesterol ester content of the particles by gas chromatography (GC). Detailed methodological procedures have been integrated to the materials and methods section of the revised manuscript. AFM analysis confirmed a discoidal shape of the rHDL1 particles with a mean height of 3.9 ± 2.6 nm and a mean width of 11.5 ± 6.3 nm, which fits well with other studies from the literature describing similar sizes for disc-shaped reconstituted rHDL particles (1-3). With an aspect ratio (AR, height / width x100) of 33,9% those particles clearly have a discoidal shape (RL Figure 1A

and Figure 1F in the revised manuscript). A representative image of an disc-shaped rHDL1 particle is shown in (RL Figure 1B and Figure 1G in the revised manuscript). Importantly, rHDL2 particles differ in size and shape from rHDL1. Those particles exhibited a mean height of $14\pm 7.7\text{nm}$ and a width of $19.7\pm 7.0\text{nm}$ when analyzed with AFM. With an AR of 71.7%, those particles acquire a more spherical character, although showing increased heterogeneity compared to the rHDL1 preparations (RL Figure 1C and Figure 1H in the revised manuscript). A representative image of an spherical-like particle is shown in (RL Figure 1D and Figure 1I in the revised manuscript). GC analysis of the rHDL preparations revealed an approximately 20-fold enrichment of cholesterol and cholesterol esters in the rHDL2 particles (RL Figure 1E and F and Figure 1J and K in the revised manuscript), which parallels the theoretical molar composition of the particles used in reconstitution experiments. Together, the authors are convinced that the conducted experiments confirm the assumed differences in shape, size and biochemical composition of the two rHDL species and support the graphical structure shown in Figure 1D of the revised manuscript. The authors added the newly acquired data to Figure 1 of the revised manuscript and updated the results section accordingly (lines 153-166).

RL Figure 1. Atomic force microscopy of rHDL particles. **A**, particle size distribution of rHDL1 particles (mean height $3.9 \pm 2.6\text{nm}$; mean width $11.5 \pm 6.3\text{nm}$, AR of 33.9%). **B**, representative deconvoluted AFM image of a discoidal rHDL1 particle. **C**, particle size distribution of rHDL2 particles (mean height $14\pm 7.7\text{nm}$, mean width

19.7±7.0nm, AR of 71.1%). **D**, representative deconvoluted AFM image of an rHDL2 particle. **E**, free cholesterol and **F**, cholesterol ester content of rHDL particles was determined using gas chromatography (GC).

- 2. (page 8) The authors show that HDL promotes apoptosis of the tumor cells. This is a very atypical effect of HDL. HDL is rather known to inhibit apoptosis. This has been shown in multiple cell types. Thus the proapoptotic effects need to be substantiated, ideally by showing the mechanism to make this different and unexpected behaviour plausible. In addition, the authors should discuss this discrepancy.*

Response. Indeed, native HDL particles possess anti-apoptotic capacity which has been shown for cell types such as macrophages and endothelial cells (4-6). Accordingly, in most of the performed experiments in this study, the pro-apoptotic effect of native HDL is small or even absent. Our data rather suggest that reconstituted, discoidal HDLs have a more pronounced capability to induce apoptosis in pancreatic cancer cells. To complement the data from viability assays and flow cytometry-based apoptosis assays (Figure 2 in the revised manuscript), we now measured Caspase 3/7 activation upon treatment of Panc02 and 6066 cells with different HDL particles. Thereby, treatment with small, disc-shaped rHDL1 particles robustly induced Caspase 3/7 in both cell lines, whereas native HDL and rHDL2 particles showed no or a diminished pro-apoptotic effect, respectively (RL Figure 2A, B and Figure 5K and L of the revised manuscript). Of note, supplementation of the cell culture medium with increasing concentrations of cholesterol decreased the pro-apoptotic effect of rHDL1 particles (RL Figure 2C). One possible explanation for the apoptosis inducing effect of rHDL1 particles might be the increased sensitivity of certain cancer cell lines to cholesterol depletion and disintegration of lipid rafts, specific plasma membrane microdomains required for the efficient concentration and activation of pro-proliferative signaling cascades such as the Ras/ERK/MAPK or AKT pathways (reviewed in (7,8)). In line, the disintegration of lipid rafts has been shown to reduce AKT phosphorylation, thereby also reducing cellular proliferation, eventually leading to the activation of pro-apoptotic mechanisms in cancer cells (9-11). By testing this hypothesis in our model, western blot analyses demonstrated the reduction of AKT phosphorylation upon rHDL1 treatment, while cholesterol-rich rHDL2 particles rather induced AKT phosphorylation (RL Figure 2D, E, F and Figure 5H, I, J of the revised manuscript). A broadly observed phenomenon in cancer cells is the constitutive activation of AKT signaling, resulting in BAD phosphorylation, thereby deactivating its proapoptotic function (reviewed in (12)). In turn, the observed reduction in AKT activation might lead to BAD-induced apoptosis of rHDL1 treated pancreatic cancer cells.

Importantly, a pro-apoptotic effect of artificial HDL species such as HDL mimetic peptides or HDL nanoparticles was also described in previous studies using different tumor models. For example, ApoA1 mimetic peptides induced apoptosis and inhibited proliferation in ovarian cancer cell lines by binding and neutralization of lysophosphatidic acid (13). Additionally, the mimetic peptide L4F reduced the tumorigenicity of H7 pancreatic cancer cells in vivo by inhibiting M2 macrophage accumulation (14). As well, in a model of B-cell lymphoma with high SR-B1 expression, HDL nanoparticles could prevent cancer progression and showed increased cholesterol efflux compared to native HDL particles (15).

RL Figure 2. Discoidal rHDL1 particles induce apoptosis and reduce AKT phosphorylation in pancreatic cancer cells. **A, B**, serum starved Panc02 or 6066 cells, respectively, were treated with indicated HDL particles (75µg/ml) in the presence of 2%FCS for 24h followed by the analysis of Caspase 3/7 activation. **C**, Caspase 3/7 activity assay upon rHDL1 treatment and supplementation with increasing concentrations of cholesterol (A-C, n=4, *p<0.05; one-way ANOVA). **D**, Western blot analysis of Panc02 cell lysates upon treatment of cells with indicated HDL particles (75µg/ml) for 18h. **E, F**, quantification of band intensities of pAKT (Thr308) and pAKT (S473) normalized to β-Actin.

Together, malignant cells might differ significantly from healthy, untransformed cells such as endothelial cells or macrophages regarding their cholesterol metabolism, lipid raft content and associated signaling networks. This might be at least one explanation for the sensitivity of cancer cells to rHDL1-mediated cholesterol depletion.

We integrated the newly acquired data to Figure 5 of the revised manuscript and updated the results section accordingly (lines 292-301). Additionally, we added a separate paragraph in the discussion regarding disintegration of lipid rafts and cancer cell apoptosis (lines 381-393).

3. (page 9). *The authors used double labeled lipoproteins to estimate the flux of cholesterol and cholesteryl esters. Ideally, the authors determine net cholesterol fluxes by measuring cholesterol mass concentrations in both cells and media. This will answer the important Question on the relative contribution of cholesterol influx and efflux by HDL.*

To determine potential changes in cellular cholesterol mass, we treated pancreatic cancer cells with different HDL species in the presence of lovastatin (5 μ M) to block compensation of cellular cholesterol pools by endogenous synthesis and analyzed cell lysates by gas chromatography (GC). Interestingly, we found decreased cholesterol levels in cells treated for 24 hours with rHDL1 and rHDL2 compared to control and nHDL treatment (RL Figure 3A). After 48h, rHDL1 treated cells showed the lowest cholesterol concentration, further substantiating our findings that rHDL1 particles efficiently deplete pancreatic cancer cells from intracellular cholesterol pools (RL Figure 3B and Figure 3F in the revised manuscript and lines 218-220).

RL Figure 3. rHDL particles reduce cellular cholesterol content. Panc02 cells were plated at a density of 0.8×10^6 cells per 6cm dish in medium containing 2% FCS, 5 μ M lovastatin and the indicated HDL particles (75 μ g/ml). Cells were harvested after **A**) 24h and **B**) 48h after 3 washes in warm PBS to remove excess lipids

from the cell culture medium. Total cholesterol content was determined by gas chromatography (GC, n=2, *p<0.05; one-way ANOVA).

Determination of cholesterol concentrations by GC in supernatants did not result in conclusive results due to the high concentration of cholesterol / lipids in FCS-containing cell culture medium and significant differences in cholesterol content of the supplemented HDL particles. The authors are aware of the fact that the low n numbers hamper statistical interpretation. However, analysis of lipid classes with GC is a rather laborious approach. Therefore, authors decided for this experimental design during the course of the revision, which fortunately resulted in reproducible and clear results.

4. (page 11): *the in vivo experiment is important and interesting. Since wild type mice and apoA1 ko mice do not differ towards tumor growth but overexpression of APOA1 or injection of rHDL reduces tumor growth in apoA1 ko mice it will be important to show the effect of APOA1 expression and rHDL in wild type mice as well. If the authors hypothesis of the need of distinct HDL particles is right, also this Intervention will reduce tumor growth. An alternative explanation is that human apoA-I but not murine apoA-I_ protects. This should be tested by according experiments in vitro*

The authors would like to thank the reviewer for this important suggestion. Unfortunately, during the course of the revision it was not possible to synthesize the required amounts of rHDL particles to perform an *in vivo* experiment with an appropriate number of animals (n=8-10). Alternatively, cell culture experiments shown in RL Figure 2 further elaborate on the mechanism of how rHDL1 particles reduce cancer cell proliferation.

To show an anti-tumor activity of AAV-delivered APOA1 in WT animals, we injected C57Bl/6 WT animals with AAV2.8 particles (those particles home specifically to the liver) encoding for murine APOA1 or control (RL Figure 4A). On the next day, mice were injected intraperitoneally with 2×10^6 Panc02 cells. After 14 days, mice were sacrificed and tumor weights were analyzed (RL Figure 4B, C). AAV-APOA1 mice showed a clear trend towards a reduction in tumor weight (p=0.09), albeit not reaching statistical significance (RL Figure 4D and Figure 5F in the revised manuscript).

RL Figure 4. AAV-APOA1 delivery to Panc02 tumor-bearing C57Bl6/J WT mice exerts a mild anti-tumor effect. **A**, C57Bl6/J WT mice were injected i.v. with either 1×10^{11} vgc of AAV-APOA1 or control on day prior to **B**, intraperitoneal inoculation of 2×10^6 Panc02 tumor cells. **C**, tumor growth was monitored for 14d. Afterwards, mice were sacrificed and tumor weight was determined (n=8/4, student's T-test).

The rather small anti-tumor effect could be explained by i) the highly aggressive growth characteristics of Panc02 cells ii) fast initiation of tumor growth paralleled by a late onset of AAV-transgene expression, iii) poor vascularization of tumor nodules, which hampers the accessibility of AAV-delivered HDL particles to tumor cells. However, the authors plan to address the therapeutic applicability of AAV-APOA1 and / or rHDL injections in future studies.

As mentioned above, data were integrated in Figure 5F of the revised manuscript and the results section was updated accordingly (lines 273-275). As well, a section regarding the limitations of the presented in vivo work including this important point was added to the discussion (lines 423-442).

5. (page 12, lines 281ff) the authors measured cholesterol efflux capacity. They only present data on cholesterol efflux capacity normalized to apoA-I levels. They must show the non-normalized data at least as well, if not only. They are more important because this reflects the physiological or pathological situation of Plasma. Moreover, by normalization, differences can be due to differences in cholesterol efflux capacity, apoA-I levels or both. The answer will determine the interpretation

This is an interesting point, and the authors agree with the reviewer that the total plasma efflux capacity is an important issue. By showing APOA1-normalized efflux, authors intended to specifically determine the efflux capacity in the HDL fraction, which is also of importance to the present study. Nevertheless, RL Figure 5 shows the non-normalized data, which have also been integrated into the revised manuscript (Figure 6D and lines 308-311 in the results section).

RL Figure 5. Non-normalized plasma efflux capacity was determined in plasma samples from a cohort of healthy volunteers and pancreatic cancer patients in late stages of their disease (n=19/19; *p<0.05; unpaired t-test). Efflux capacity was calculated as the percentage of ^3H cholesterol tracer measured in the supernatant in relation to the amount of intracellular radiotracer after incubation of cells with acceptor plasma samples for 8h. PBS control acceptor samples were used to identify unspecific ^3H cholesterol transfer into the cell culture medium.

The reduced plasma efflux capacity observed in samples from pancreatic cancer patients might result from chronic inflammatory processes that manifest in the tumor context, including the secretion of pro-inflammatory cytokines from the tumor and the expansion and activation of diverse immune cell populations such as macrophages, MDSC, T cells and NK cells. Chronic inflammation is a driving force for the generation of dysfunctional HDL particles. For example, the association of the acute phase protein SAA1 with plasma HDLs has been shown to reduce the efflux capacity of HDL (16), which is corroborated by our findings (Figure 6F and G in the revised manuscript). Tumor-mediated remodeling of the HDL particle composition might therefore lead to a decreased efflux capacity, which in turn could contribute to increased tumor malignancy. As well, dysfunctional HDLs associated with metabolic diseases such as obesity or diabetes (16,17) likely represent tumor promoting factors, thereby contributing to the increased risk of such patients to develop cancer.

6. (page 13). *the Kaplan Meier analysis is interesting. However, in the context of the present study, it will be most interesting to see if SCARB1 and ABCA1 expression is associated with survival. If not, the interpretation will have to be substantially changed.*

The TCGA database provides a small cohort of PDAC patients, which has been used for Kaplan Meier analysis. When correlating SCARB1 and ABCA1 expression with overall survival (OS) of patients, no significant differences are apparent (RL Figure 6A and B). This might be due to several reasons. First, the collective of patients is rather small (n=176 samples), which might hamper proper interpretation. Second, and according to our findings, expression of SCARB1 and ABCA1 might not per se determine malignancy, but rather their ligands (different HDL particle composition and maybe also LDL particles).

RL Figure 6. Kaplan Meier analysis correlating **A**, SCARB1 and **B**, ABCA1 expression with overall patient survival in the GDC-TCGA pancreatic cancer cohort (n=176, expression data are shown as $\log_2(\text{fpkm-}uq+1)$ transformed data, log-rank test, *p<0.05, UCSC Xena).

Patient specific plasma lipid / lipoprotein profiles might differentially regulate the supply of cancer cells with cholesterol, thereby influencing cancer cell growth characteristics. Patient lipid and lipoprotein profiles might therefore significantly add to cancer progression. Interestingly, obesity and the metabolic syndrome are well recognized risk factors for the development of tumor malignancies of various types (18).

Kaplan Meier analysis correlating patient overall survival with SCARB1 and ABCA1 expression is now shown in Supplementary Figure 4A and B of the revised manuscript and the results section was updated accordingly (lines 319-324). Combinatorial approaches to manipulate efflux receptor expression on cancer cells and rHDL delivery are discussed (lines 366-380).

7. *(page 14) lines 324 ff and page 15 line 1ff. The authors discuss prebeta1 HDL. Thereby they claim that rHDL1 is an analog of prebeta1 HDL. This is principally wrong, because prebeta1 HDL corresponds to lipid-free or poor apoA-I. rHDL1 contains 100 moles of PC per mole of apoA-I and has alpha rather than prebeta mobility as also shown by the authors in Figure 1G. Looking to this figure one sees that rHDL1 migrates faster rather than slower than native HDL and is a prealpha- rather than a prebeta-particle. In fact, rHDL1 is an analogue of the ABCA1 product (rather than educt) and the LCAT educt.*

The authors apologize for the imprecise HDL particle nomenclature regarding pre- β HDL and agrees with the reviewer that rHDL1 most likely represents an analogue of the ABCA1 product. In the discussion, the authors compared rHDL1 to be an analog of pre- β particles (not pre- β 1 as stated by the reviewer), as a substantial amount of literature describes small, discoidal HDL particles (basically the ABCA1 product) as pre- β HDLs (see references (19-22) amongst others). However, the authors are also aware of the fact that the analysis of HDL particles with agarose gel electrophoresis as applied in the present manuscript does not provide the resolution to draw conclusive interpretations regarding the definitive particle type. Therefore, and according to our AFM analyses (RL Figure 1), which clearly identify rHDL1 particles as disc-shaped, but also show a significant heterogeneity regarding rHDL2 shape and size, the author now changed the particle nomenclature to discoidal (for rHDL1) and spherical-like particles (for rHDL2), thereby for the most part avoiding the terminology of β or α migrating particles.

10. page 22: The authors use parametric statistical Tests without showing normality. The low numbers of in vitro experiments make non-parametric tests preferable anyway. More and more journals request this, for example if $N < 6$.

Authors apologize for this missing information and provide a more detailed description of the used statistical methods in the revised manuscript. To compare the means of two groups, an unpaired, two-tailed student's t-test was used. Pairwise comparison testing in experiments with more than two groups was performed using one-way ANOVA followed by Tukey post hoc test for multiple comparisons. Pairwise comparisons of tumor growth kinetics were performed using two-way ANOVA followed by bonferroni post hoc test for multiple comparisons. In all cases statistical significance was assumed when $p < 0.05$.

We now performed normality testing wherever possible, however, Anderson Darling as well as D'Agostino and Pearson normality test require cohort sizes of a minimum of $n=8$ values. As many experiments/groups contained group sizes of $n < 8$, analyzing Gaussian distribution was performed with the Shapiro-wilk test wherever possible. Of note, our statistical approach is in accordance with other manuscripts published in Life Science Alliance (23-25). Accordingly, we updated the Statistics section in the Material and Methods part of the revised manuscript (see lines 634-641).

REVIEWER B

- Figure 1 (first part) makes the case that cholesterol depletion inhibits tumor growth in vitro. However, conditions for cholesterol depletion in Fig 1A are not clear. Why authors add mevalonate (which is a cholesterol precursor and can potentially support endogenous biosynthesis)? Could they do that with cyclodextrins?

The settings used here to induce cholesterol depletion combine 2% lipoprotein deficient serum (LPDS) and lovastatin (5 μ M) to block endogenous cholesterol synthesis. This combination is a rather harsh treatment condition for cells, which does usually not allow prolonged incubation periods as applied in this experimental setting. Therefore, a defined concentration of mevalonate (100 μ M) is added to the cell culture medium to supply the cells with minimal cholesterol building blocks. Depletion of cellular cholesterol pools by LPDS and statins has been described initially in papers by Brown and Goldstein (26,27). The addition of mevalonate is frequently used to deplete cells from cholesterol depots while giving them the minimal amount that is required to survive a certain timeframe (28,29). At least in our opinion, this protocol represents a more suitable method to study cholesterol depletion compared to cyclodextrins. These cyclodextrins (for example methyl- β -cyclodextrin (MBCD)) deplete cells very efficiently from cholesterol in as little as 2-6 hours, show potent cytotoxicity and induce apoptosis in a wide range of cell lines, especially when used at high concentrations >5mM (9,30-32).

Nevertheless, we performed viability assays with increasing concentrations of MBCD. While low concentrations of MBCD (0.5mM) did not show changes in cellular viability, the addition of 2mM almost completely abolished viability which was also accompanied by a loss of cellular integrity (RL Figure 7A). Interestingly, when MBCD was used in apoptosis assays (24h incubation period), we were not able to observe caspase 3/7 activation. In contrast, rHDL1 treatment induced a solid increase in Caspase 3/7 activity (RL Figure 7B).

RL Figure 7. Panc02 cells are sensitive to methyl- β -cyclodextrin (MBCD) at high concentrations. Panc02 cells were serum-starved overnight and seeded the next day at a density of 1×10^4 cells / 100 μ l in a 96 well plate in the presence of either 0.5mM MBCD, 2mM MBCD or rHDL1 (75 μ g/ml) in medium containing 2%FCS. A,

viability of Panc02 cells was assessed by WST1 assay. B, the apoptosis inducing capacity of MBCD was measured by a Caspase 3/7 activation assay (n=4 for each treatment condition).

Together, our results indicate that MBCD, especially at higher concentrations, induces cell death in Panc02 cells independently of caspase 3/7-mediated apoptosis at least in the here applied conditions. In the opinion of the authors, the usage of MBCD at concentrations >2mM therefore represents a harsh and unspecific method to kill (cancer) cells of almost any origin by highly efficient extraction of cellular cholesterol.

- Figure 2: HDL composition affects viability. Excellent experimentation. No major issues, although I would have preferred analysis of SREB activation rather qPCR of downstream targets. But data look OK. Please, clarify that data in F thru H express fold change over CONTROL.

The authors would like to thank the reviewer for the positive feedback on the experimental setup. As assumed, the data are expressed as fold change over PBS-treated control cells. To make this issue more clear, we specifically state 'data are expressed as fold change over PBS control in the Figure legend of Figure 2 (see lines 685-686 in the revised manuscript).

- Figure 4 explains the HDL composition dictates cholesterol flux direction and impacts tumor cell viability. No issues, authors are commended for clever strategy. However, definitive proof for ABCA1 role would require knockdown experiments, so maybe authors want to comment that.

The authors agree with the reviewer that knockdown of ABCA1 would have been an appropriate choice to proof the proposed role of the protein in this regulatory network. One issue that makes this approach unfortunately hardly feasible is the low endogenous expression levels of ABCA1 in the applied pancreatic cancer cell lines. Although sensitive towards LXR agonists that efficiently increase ABCA1 expression in Panc02 cells, baseline ABCA1 protein levels are hardly detectable with e.g. Western blotting (see Figure 4A of the revised manuscript). Therefore, authors decided against knockdown experiments, as the knockdown itself would have been difficult to evaluate. However, the presented data that activation of ABCA1 with LXR agonists further exacerbates the proapoptotic effect of cholesterol depleting rHDL particles raises some interesting questions. For example, are there differences in the binding affinity of distinct HDL subpopulations for ABCA1 or SR-B1 expressed on cancer cells? Would it be possible to identify the subcellular cholesterol pools that are primarily depleted by the combination of LXR agonists and rHDL1 particles? As the pro-apoptotic effect by Caspase 3/7 activation is already detectable after 24 hours, it might be that disintegration of lipid rafts, which are important for maintaining pro-proliferative

signaling pathways of cancer cells, might be a central mechanism. Eventually, from a translational point, it would be interesting to test whether a combination of LXR agonists with rHDL1 particles might act synergistically to block tumor formation *in vivo*.

The authors briefly discuss some aspects of the current knowledge and therapeutic implications of LXR agonists in tumor biology (366-372).

- Figure 5 describe therapeutic potential of rHDL. Results are neat and experiments well performed but I don't understand the observed differences between WT, ABCA1 KO and AAV-reconstituted mice. APO levels appear similar between WT and AAV, so why the difference in tumor growth? Authors do state native HDL appear irrelevant for tumor growth, so where's the change? AAV-transduce mice produce different HDL? Shouldn't the authors show data for that? Is that somehow already known? I think the authors should either comment a bit more on that or show that HDL particles are indeed different in the two cohorts.

The fact that AAV-delivered HDL particles were not analyzed regarding their composition is definitely a shortcoming of the study. Initially, AAV-delivery of APOA1 was thought to increase HDL levels in APOA1 KO mice, thereby increasing cholesterol efflux, eventually decreasing tumor growth. However, our observation that WT and APOA1 KO mice do not differ in tumor burden definitely raises the questions of particle composition, turnover and functionality of AAV-delivered HDL. While revising this manuscript, an experiment was conducted that explored a potential anti-tumor activity of AAV-delivered APOA1 in WT animals as suggested by reviewer A. However, it was not possible to analyze those particles regarding their structural- and / or biochemical composition due to time limitations and technical reasons to discriminate endogenous from AAV-delivered APOA1 / HDL. Interestingly, a study by Lebherz and Rader showed that the hepatic expression of full length human APOA1 as well as the APOA1 Milano variant using AAV2.8 particles reduced HDL cholesterol levels in WT mice while reducing atherosclerotic burden (33). Although the detailed mechanism of this observation remained ambiguous, the authors hypothesize that HDL particle catabolism, and therefore HDL-mediated cholesterol efflux might increase (33). Of note, carriers of the APOA1 Milano variant (SNP Arg173Cys) also exhibit reduced HDL cholesterol levels, which is associated with atheroprotection. As well, therapeutic administration of APOA1 Milano shows significant anti-atherosclerotic activity (34). One hint that HDL catabolism might be increased in tumor bearing APOA1 KO mice reconstituted with AAV-APOA1 particles is reflected by low HDL cholesterol levels compared to WT animals albeit the robust expression of APOA1 in those animals (Figure 5C of the revised manuscript). To definitely proof this hypothesis, HDL turnover / *in vivo* reverse cholesterol transport studies (35) with ³H cholesterol labeled tumor cells would be required to analyze

the potential increase in AAV-delivered HDL catabolism. The authors now added a separate paragraph to the discussion section elaborating on this limitation of the study (lines 423-442).

- Figure 6 shows cholesterol parameters in PDAC patients. I couldn't find description of the patients cohort anywhere, nor compliance with ethical requirements. Please add that. Not clear how they calculate cholesterol efflux % in D.

Ethical requirements are stated in the Methods section of the revised manuscript (lines 596-601). We now also added the main patient characteristics to this section. Plasma samples from 20 pancreatic cancer patients (10 female, 10 male, mean age 69 of years, mean BMI of 24.6) in advanced stages of their disease were collected at the department of Medicine I at the Medical University of Vienna. Human blood samples were collected into pre-coated EDTA tubes under informed consent of patients as well as approval by and strict adherence to institutional guidelines of the Medical University of Vienna and the Declaration of Helsinki (Ethik Votum 1035/2020).

Cholesterol efflux capacity was calculated as the percentage of ^3H cholesterol tracer measured in the supernatant in relation to the amount of intracellular radiotracer after incubation of cells with acceptor plasma samples for 8h. PBS control acceptor samples were used to identify unspecific ^3H cholesterol transfer into the cell culture medium. For normalization, APOA1 plasma levels were determined by ELISA and cholesterol efflux values were divided by those values.

REFERENCES

1. Lerch, P. G., Fortsch, V., Hodler, G., and Bolli, R. (1996) Production and characterization of a reconstituted high density lipoprotein for therapeutic applications. *Vox Sang* **71**, 155-164
2. Rye, K. A. (1990) Interaction of apolipoprotein A-II with recombinant HDL containing egg phosphatidylcholine, unesterified cholesterol and apolipoprotein A-I. *Biochim Biophys Acta* **1042**, 227-236
3. Sevugan Chetty, P., Mayne, L., Kan, Z. Y., Lund-Katz, S., Englander, S. W., and Phillips, M. C. (2012) Apolipoprotein A-I helical structure and stability in discoidal high-density lipoprotein (HDL) particles by hydrogen exchange and mass spectrometry. *Proc Natl Acad Sci U S A* **109**, 11687-11692
4. Feuerborn, R., Becker, S., Poti, F., Nagel, P., Brodde, M., Schmidt, H., Christoffersen, C., Ceglarek, U., Burkhardt, R., and Nofer, J. R. (2017) High density lipoprotein (HDL)-associated sphingosine 1-phosphate (S1P) inhibits macrophage apoptosis by stimulating STAT3 activity and survivin expression. *Atherosclerosis* **257**, 29-37
5. Nofer, J. R., Levkau, B., Wolinska, I., Junker, R., Fobker, M., von Eckardstein, A., Seedorf, U., and Assmann, G. (2001) Suppression of endothelial cell apoptosis by high

- density lipoproteins (HDL) and HDL-associated lysosphingolipids. *J Biol Chem* **276**, 34480-34485
6. Terasaka, N., Wang, N., Yvan-Charvet, L., and Tall, A. R. (2007) High-density lipoprotein protects macrophages from oxidized low-density lipoprotein-induced apoptosis by promoting efflux of 7-ketocholesterol via ABCG1. *Proc Natl Acad Sci U S A* **104**, 15093-15098
 7. Mollinedo, F., and Gajate, C. (2020) Lipid rafts as signaling hubs in cancer cell survival/death and invasion: implications in tumor progression and therapy: Thematic Review Series: Biology of Lipid Rafts. *J Lipid Res* **61**, 611-635
 8. Simons, K., and Toomre, D. (2000) Lipid rafts and signal transduction. *Nat Rev Mol Cell Biol* **1**, 31-39
 9. Calay, D., Vind-Kezunovic, D., Frankart, A., Lambert, S., Poumay, Y., and Gniadecki, R. (2010) Inhibition of Akt signaling by exclusion from lipid rafts in normal and transformed epidermal keratinocytes. *J Invest Dermatol* **130**, 1136-1145
 10. Krycer, J. R., Sharpe, L. J., Luu, W., and Brown, A. J. (2010) The Akt-SREBP nexus: cell signaling meets lipid metabolism. *Trends Endocrinol Metab* **21**, 268-276
 11. Zhuang, L., Lin, J., Lu, M. L., Solomon, K. R., and Freeman, M. R. (2002) Cholesterol-rich lipid rafts mediate akt-regulated survival in prostate cancer cells. *Cancer Res* **62**, 2227-2231
 12. Matsuura, K., Canfield, K., Feng, W., and Kurokawa, M. (2016) Metabolic Regulation of Apoptosis in Cancer. *Int Rev Cell Mol Biol* **327**, 43-87
 13. Su, F., Grijalva, V., Navab, K., Ganapathy, E., Meriwether, D., Imaizumi, S., Navab, M., Fogelman, A. M., Reddy, S. T., and Farias-Eisner, R. (2012) HDL mimetics inhibit tumor development in both induced and spontaneous mouse models of colon cancer. *Mol Cancer Ther* **11**, 1311-1319
 14. Peng, M., Zhang, Q., Cheng, Y., Fu, S., Yang, H., Guo, X., Zhang, J., Wang, L., Zhang, L., Xue, Z., Li, Y., Da, Y., Yao, Z., Qiao, L., and Zhang, R. (2017) Apolipoprotein A-I mimetic peptide 4F suppresses tumor-associated macrophages and pancreatic cancer progression. *Oncotarget* **8**, 99693-99706
 15. Yang, S., Damiano, M. G., Zhang, H., Tripathy, S., Luthi, A. J., Rink, J. S., Ugolkov, A. V., Singh, A. T., Dave, S. S., Gordon, L. I., and Thaxton, C. S. (2013) Biomimetic, synthetic HDL nanostructures for lymphoma. *Proc Natl Acad Sci U S A* **110**, 2511-2516
 16. Vaisar, T., Tang, C., Babenko, I., Hutchins, P., Wimberger, J., Suffredini, A. F., and Heinecke, J. W. (2015) Inflammatory remodeling of the HDL proteome impairs cholesterol efflux capacity. *J Lipid Res* **56**, 1519-1530
 17. Talbot, C. P. J., Plat, J., Joris, P. J., Konings, M., Kusters, Y., Schalkwijk, C. G., Ritsch, A., and Mensink, R. P. (2018) HDL cholesterol efflux capacity and cholesteryl ester transfer are associated with body mass, but are not changed by diet-induced weight loss: A randomized trial in abdominally obese men. *Atherosclerosis* **274**, 23-28
 18. Font-Burgada, J., Sun, B., and Karin, M. (2016) Obesity and Cancer: The Oil that Feeds the Flame. *Cell Metab* **23**, 48-62
 19. Camont, L., Chapman, M. J., and Kontush, A. (2011) Biological activities of HDL subpopulations and their relevance to cardiovascular disease. *Trends Mol Med* **17**, 594-603
 20. Favari, E., Calabresi, L., Adorni, M. P., Jessup, W., Simonelli, S., Franceschini, G., and Bernini, F. (2009) Small discoidal pre-beta1 HDL particles are efficient acceptors of cell cholesterol via ABCA1 and ABCG1. *Biochemistry* **48**, 11067-11074

21. Lewis, G. F., and Rader, D. J. (2005) New insights into the regulation of HDL metabolism and reverse cholesterol transport. *Circ Res* **96**, 1221-1232
22. Rye, K. A., and Barter, P. J. (2004) Formation and metabolism of prebeta-migrating, lipid-poor apolipoprotein A-I. *Arterioscler Thromb Vasc Biol* **24**, 421-428
23. Adachi, Y., Masuda, M., Sakakibara, I., Uchida, T., Niida, Y., Mori, Y., Kamei, Y., Okumura, Y., Ohminami, H., Ohnishi, K., Yamanaka-Okumura, H., Nikawa, T., and Taketani, Y. (2022) All-trans retinoic acid changes muscle fiber type via increasing GADD34 dependent on MAPK signal. *Life Sci Alliance* **5**
24. Vitner, E. B., Avraham, R., Politi, B., Melamed, S., and Israely, T. (2022) Elevation in sphingolipid upon SARS-CoV-2 infection: possible implications for COVID-19 pathology. *Life Sci Alliance* **5**
25. Xie, M., Chia, R. H., Li, D., Teo, F. X., Krueger, C., and Sabapathy, K. (2021) Functional interaction between macrophages and hepatocytes dictate the outcome of liver fibrosis. *Life Sci Alliance* **4**
26. Brown, M. S., Dana, S. E., and Goldstein, J. L. (1973) Regulation of 3-hydroxy-3-methylglutaryl coenzyme A reductase activity in human fibroblasts by lipoproteins. *Proc Natl Acad Sci U S A* **70**, 2162-2166
27. Brown, M. S., and Goldstein, J. L. (1980) Multivalent feedback regulation of HMG CoA reductase, a control mechanism coordinating isoprenoid synthesis and cell growth. *J Lipid Res* **21**, 505-517
28. Loregger, A., Cook, E. C., Nelson, J. K., Moeton, M., Sharpe, L. J., Engberg, S., Karimova, M., Lambert, G., Brown, A. J., and Zelcer, N. (2016) A MARCH6 and IDOL E3 Ubiquitin Ligase Circuit Uncouples Cholesterol Synthesis from Lipoprotein Uptake in Hepatocytes. *Mol Cell Biol* **36**, 285-294
29. Trinh, M. N., Brown, M. S., Goldstein, J. L., Han, J., Vale, G., McDonald, J. G., Seemann, J., Mendell, J. T., and Lu, F. (2020) Last step in the path of LDL cholesterol from lysosome to plasma membrane to ER is governed by phosphatidylserine. *Proc Natl Acad Sci U S A* **117**, 18521-18529
30. Li, Y. C., Park, M. J., Ye, S. K., Kim, C. W., and Kim, Y. N. (2006) Elevated levels of cholesterol-rich lipid rafts in cancer cells are correlated with apoptosis sensitivity induced by cholesterol-depleting agents. *Am J Pathol* **168**, 1107-1118; quiz 1404-1105
31. Onodera, R., Motoyama, K., Okamatsu, A., Higashi, T., Kariya, R., Okada, S., and Arima, H. (2013) Involvement of cholesterol depletion from lipid rafts in apoptosis induced by methyl-beta-cyclodextrin. *Int J Pharm* **452**, 116-123
32. Yamaguchi, R., Perkins, G., and Hirota, K. (2015) Targeting cholesterol with beta-cyclodextrin sensitizes cancer cells for apoptosis. *FEBS Lett* **589**, 4097-4105
33. Leberer, C., Sanmiguel, J., Wilson, J. M., and Rader, D. J. (2007) Gene transfer of wild-type apoA-I and apoA-I Milano reduce atherosclerosis to a similar extent. *Cardiovasc Diabetol* **6**, 15
34. Chiesa, G., and Sirtori, C. R. (2003) Apolipoprotein A-I(Milano): current perspectives. *Curr Opin Lipidol* **14**, 159-163
35. Bartelt, A., John, C., Schaltenberg, N., Berbee, J. F. P., Worthmann, A., Cherradi, M. L., Schlein, C., Piepenburg, J., Boon, M. R., Rinninger, F., Heine, M., Toedter, K., Niemeier, A., Nilsson, S. K., Fischer, M., Wijers, S. L., van Marken Lichtenbelt, W., Scheja, L., Rensen, P. C. N., and Heeren, J. (2017) Thermogenic adipocytes promote HDL turnover and reverse cholesterol transport. *Nat Commun* **8**, 15010

April 25, 2022

RE: Life Science Alliance Manuscript #LSA-2021-01317-TR

Dr. Raimund Oberle
Medical University of Vienna
Institute of Medical Chemistry, Center for Pathobiochemistry and Genetics
Währingerstraße 10
Vienna, Vienna 1090
Austria

Dear Dr. Oberle,

Thank you for submitting your revised manuscript entitled "The HDL particle composition determines its anti-tumor activity in pancreatic cancer". We would be happy to publish your paper in Life Science Alliance pending final revisions necessary to meet our formatting guidelines.

- please consult our manuscript preparation guidelines <https://www.life-science-alliance.org/manuscript-prep> and make sure your manuscript sections are in the correct order
- please upload your supplementary figure files as single files and add the figure legends for the supplementary figures to the main manuscript file, directly under the main figure legends
- please upload your tables as editable doc or excel files
- please add the Twitter handle of your host institute/organization as well as your own or/and one of the authors in our system
- please add the author contributions to your main manuscript text
- please use the [10 author names, et al.] format in your references (i.e. limit the author names to the first 10)
- please add a callout for Table S1 in your main manuscript text
- you may upload Figure 7 as a Graphical Abstract instead, if you prefer. Details can be found here: <https://www.life-science-alliance.org/manuscript-prep#presentation>

A. FINAL FILES:

B. MANUSCRIPT ORGANIZATION AND FORMATTING:

Sincerely,

Reviewer #1 (Comments to the Authors (Required)):

I thank the authors for taking my criticisms as constructive advice. I am very satisfied with the rebuttal and revision and have no further suggestions for improvement

Reviewer #2 (Comments to the Authors (Required)):

The authors did address all my concerns, adding valuable experiments and significant results to the original manuscript. New data improved the manuscript.
Congratulations to all the authors

May 3, 2022

RE: Life Science Alliance Manuscript #LSA-2021-01317-TRR

Dr. Raimund Oberle
Medical University of Vienna
Institute of Medical Chemistry, Center for Pathobiochemistry and Genetics
Währingerstraße 10
Vienna, Vienna 1090
Austria

Dear Dr. Oberle,

Thank you for submitting your Research Article entitled "The HDL particle composition determines its anti-tumor activity in pancreatic cancer". It is a pleasure to let you know that your manuscript is now accepted for publication in Life Science Alliance. Congratulations on this interesting work.

DISTRIBUTION OF MATERIALS:

Again, congratulations on a very nice paper. I hope you found the review process to be constructive and are pleased with how the manuscript was handled editorially. We look forward to future exciting submissions from your lab.

Sincerely,
